# Sulcal organization in the medial frontal cortex provides insights into primate brain evolution

Céline Amiez [1,10], Jérôme Sallet [2,10], William D. Hopkins[3], Adrien Meguerditchian [4,5,6], Fadila Hadj-Bouziane [7], Suliann Ben Hamed [8], Charles R.E. Wilson [1], Emmanuel Procyk [1] & Michael Petrides[9]

Although the relative expansion of the frontal cortex in primate evolution is generally accepted, the nature of the human uniqueness, if any, and between-species anatomo-functional comparisons of the frontal areas remain controversial. To provide a novel interpretation of the evolution of primate brains, sulcal morphological variability of the medial frontal cortex was assessed in Old World monkeys (macaque/baboon) and Hominoidea (chimpanzee/human). We show that both Hominoidea possess a paracingulate sulcus, which was previously thought to be unique to the human brain and linked to higher cognitive functions, such as mentalizing. Also, we show systematic sulcal morphological organization of the medial frontal cortex that can be traced from Old World monkeys to Hominoidea species, demonstrating an evolutionarily conserved organizational principle. These data provide a new framework to compare sulcal morphology, cytoarchitectonic areal distribution, connectivity, and function across the primate order, leading to clear predictions about how other primate brains might be anatomo-functionally organized.

[1] Univ Lyon, Université Lyon 1, Inserm, Stem Cell and Brain Research Institute U1208, 69500 Bron, France. [2] Wellcome Integrative Neuroimaging Centre, Department of Experimental Psychology, University of Oxford, Oxford OX1 3SR, UK. [3] Department of Comparative Medicine, University of Texas MD Anderson Cancer Center, Bastrop, TX 78602, USA. [4] Laboratoire de Psychologie Cognitive, UMR7290, Université Aix-Marseille, CNRS, 13331 Marseille, France. [5] Station de Primatologie CNRS, UPS846, 13790 Rousset, France. [6] Brain & Language Research Institute, Université Aix-Marseille, CNRS, 13604 Aix-en-Provence, France. [7] Integrative Multisensory Perception Action & Cognition Team (ImpAct), INSERM U1028, CNRS UMR5292, Lyon Neuroscience Research Center (CRNL), University of Lyon 1, 69500 Lyon, France. [8] Institut des Sciences Cognitives Marc Jeannerod, UMR5229, CNRS-Université Claude Bernard Lyon I, 69675 Bron, France. [9] Montreal Neurological Institute, Department of Neurology and Neurosurgery and Department of Psychology, McGill University, Montreal, QC H3A 2B4, Canada. [10] These authors contributed equally: Céline Amiez, Jérôme Sallet. Correspondence and requests for materials should be addressed to C.A. (email: celine.amiez@inserm.fr)

How is the human brain unique? Although the relative expansion of the frontal cortex in primate evolution is generally accepted, the nature of the changes that occur remains controversial. Neuroanatomy offers a window into the evolution of brain circuits and their functions. At a macroscopic level, a large body of literature has emphasized the link between the extent of gyrification, the rapid expansion of the cerebral cortex, and the complexity of the computational processing performed in a given brain[1]. Several hypotheses have been proposed to explain the origin of sulci and gyri: genetic control[1], cortical growth[2], tension of white matter cortico-cortical axons[3], cortico-thalamic axons[4], or simply mechanical instability in a soft tissue growing non-uniformly[5]. Although important, previous discussions of cortical gyrification have not considered another major dimension of sulcal pattern organization, i.e., its variability. Inter-individual variability of traits is at the basis of many evolutionary genetic studies[6,7]. Thus, the primary goal of the present study was to investigate variability in sulcal phenotypes in four key primate populations to provide new evidence and understanding of the evolution of primate frontal cortex.

Sulcal patterns follow a precise topographical organization[8]. It is important to note that sulcal organization is not random despite the presence of significant inter-hemispheric and inter-subject variability. A straightforward example is the location of the central sulcus. Although its shape, length, and depth may vary across hemispheres and individuals, it is always present and systematically located at the same strategic antero-posterior location. Importantly, this is not a human specific feature as it is observed in all primates[9]. Although the origin of gyrification is not well understood, three types of sulci can be identified in primates, based on their appearance during gestation. Primary sulci, which appear first during gestation (e.g., central sulcus, cingulate sulcus –CGS–)[10,11], are present in all hemispheres and in all individuals. By contrast, the probability of observing secondary/tertiary sulci is variable. For example, the paracingulate sulcus (PCGS) is present only in about 70% of subjects at least in one hemisphere[12–14].

Importantly, several lines of evidence indicate that many primary sulci are limiting sulci between cytoarchitectonic areas. For instance, the central sulcus is the limiting sulcus between the primary motor cortex (area 4) which occupies the anterior bank of this sulcus and extends for a variable distance on the precentral gyrus and the primary somatosensory cortex (area 3) that lies on its posterior bank[15–17]. Although there is some inter-individual variability in the relation of the precise border of these areas and macrostructural features across individual brains[18], the central sulcus remains a dividing line between the motor cortical region, anteriorly, and the somatosensory cortical region, posteriorly[17,19]. Several studies performing single subject analysis have also demonstrated that the organization of sulci and their variability is pertinent to prediction of the location of functional areas in various brain regions, e.g., refs. [20,21]. Collectively, these findings strongly suggest that sulcal organization in primates is not random, but rather has anatomo-functional relevance that is likely associated with the evolution of increasingly sophisticated sensory, motor, and cognitive functions. To what extent sulcal morphology across primate brains can provide insights into the anatomo-functional organization of brains in higher primates and brain evolution in the primate order remains poorly understood.

The aim of the present study is to determine how the medial frontal cortex (MFC) sulcal organization has evolved through the primate order. For instance, one study[13] has shown that the cytoarchitectonic organization of the cingulate cortical region is modulated, depending on the presence/absence of the PCGS. This cingulate/paracingulate region is often neglected in comparative studies[22], despite its key role in cognitive processing often thought to be unique to the human brain, such as mentalizing or counterfactual thinking[23]. Whether the MFC is anatomo-functionally comparable in the human and macaque brains remains a subject for debate[24,25]. Based on neuroimaging anatomical scans, we investigate the sulcal morphological variability in Old-world monkeys (80 rhesus monkeys [*Macaca mulatta*] and 88 baboons [*Papio papio*]) and Hominoidea (225 chimpanzees [*Pan troglodytes*] and 197 humans [*Homo sapiens*]) to identify potential specific characteristics of the human MFC.

The analysis of the sulcal organization in the MFC of primates provides critical new evidence of how the frontal cortex evolved within the primate order. We here show that both Hominoidea possess a PCGS, which was previously thought to be unique to the human brain and linked to higher cognitive functions. We also confirm the expansion of the most rostral part of the MFC in humans. But overall, we reveal systematic sulcal morphological organization of the MFC that can be traced from Old-world monkeys to Hominoidea, demonstrating an evolutionarily conserved organizational principle.

## Results

**The paracingulate sulcus (PCGS): an innovation of the brain in Hominoidea.** A major finding of the present analysis was the demonstration that a PCGS can be observed, at least in one hemisphere, in 70.1% of human brains and 33.8% of chimpanzee brains, but not in baboon and macaque brains (dependent variable: PCGS present (0/1), main effect species: $\chi^2 = 242.18$, df = 3, $p$-value = 2.2e−16, logistic regression, GLM fitted using an adjusted-score approach to bias reduction) (Fig. 1). A post-hoc Tukey test demonstrated that the probability of occurrence of a PCGS is significantly decreased from human to chimpanzee brains (estimate = 1.47562, std. error = 0.20917, $z$-value = 7.054, $p$-value < 0.001) (Fig. 1e). Interestingly, human and chimpanzee brains differed with respect to asymmetries in the PCGS. In human brains, the PCGS was present in the left hemisphere in 35% of subjects, in the right hemisphere in 12.7% of subjects, and in both hemispheres in 22.8% of subjects, a distribution that differed significantly (dependent variable: PCGS present (0/1), main effect of left versus right hemispheres: $\chi^2 = 19.912$, df = 1, $p$-value = 8.109e−06, logistic regression) (Fig. 1e). In contrast, in the chimpanzee, the PCGS was present in the left hemisphere in 12.9% of subjects, in the right hemisphere in 14.2% of subjects, and in both hemispheres in 6.7% of subjects, a distribution that did not differ significantly (dependent variable: PCGS present (0/1), main effect of left versus right hemispheres: $\chi^2 = 0.12399$, df = 1, $p$-value = 0.7247).

The presence/absence of an intralimbic sulcus (ILS), which is a shallow sulcus coursing parallel and ventral to the cingulate sulcus, was also examined in the four primate samples. The probability of observing an ILS in at least one hemisphere differed between species (proportions: 27%, 18.7%, 12.5%, and 11.3% of human, chimpanzee, baboon, and macaque brains, respectively; logistic regression, dependent variable: ILS present (0/1), main effect of species: $\chi^2 = 18.993$, df = 3, $p$-value = 0.0002743, Fig. 2a). Post-hoc Tukey analysis showed that the presence of an ILS in human brains was significantly higher than in all other primate species. This analysis revealed no significant differences in the presence of the ILS between the three non-human primate species (chimpanzee versus baboon: $p$-value = 0.5502, chimpanzee versus macaque: $p$-value = 0.4176, baboon versus macaque: $p$-value = 0.9942).

We also tested for asymmetries in the presence of ILS within each species by assessing whether the ILS was more frequent in one hemisphere compared to the other. No interhemispheric differences were found in the human brain (dependent variable:

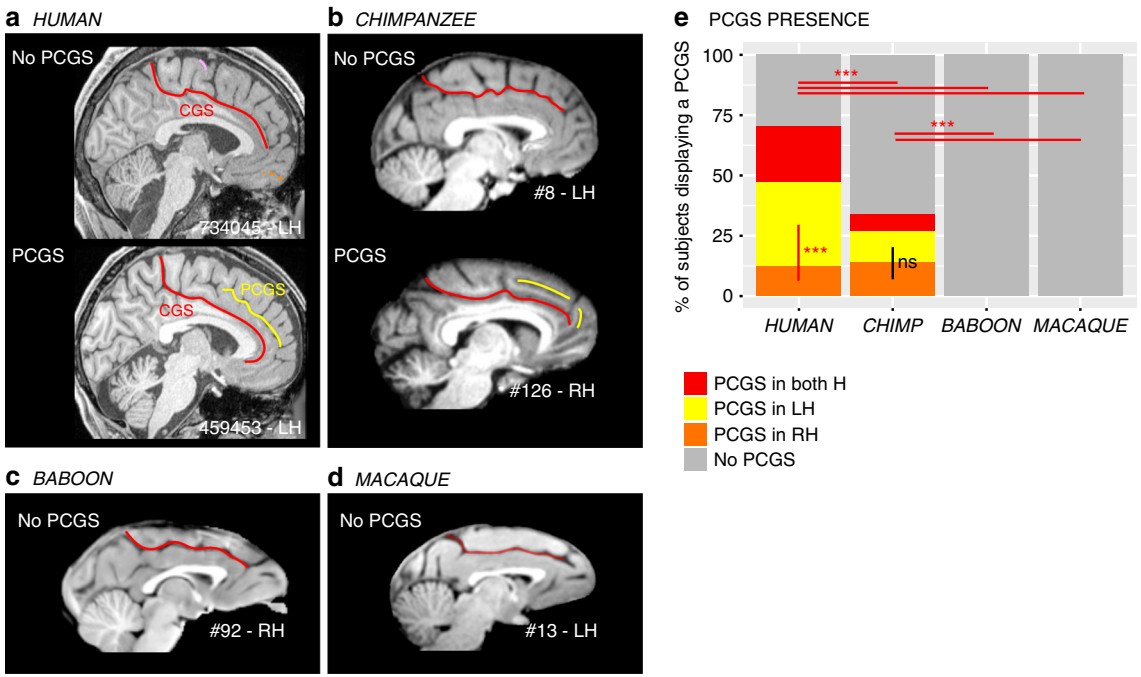

**Fig. 1** Presence of the paracingulate sulcus (PCGS) across primates. The location of the cingulate sulcus (CGS) is shown in red in typical hemispheres displaying no PCGS and in those displaying a PCGS in human (**a**) and chimpanzee (**b**), as well as in typical hemispheres of baboon (**c**) and macaque (**d**) brains. The location of the PCGS is shown in yellow in typical hemispheres displaying a PCGS in the human (**a**) and chimpanzee brains (**b**). A PCGS is observed in both human and chimpanzee brains, but not in baboon and macaque brains (**e**). The probability of occurrence of a PCGS decreased from Human to chimpanzee. The PCGS is lateralized in the left hemisphere in human, but not in chimpanzee brains. Statistics: *$p < 0.05$, **$p < 0.01$, ***$p < 0.001$, ns non-significant. Source data are provided as a Source Data file

ILS present (0/1), main effect hemispheres (left/right): $\chi^2 = 0.68999$, df = 1, $p$-value = 0.4062 (ns)), the baboon brain ($\chi^2 = 3.3591$, df = 1, $p$-value = 0.06684 (ns)), or the macaque brain ($\chi^2 = 0.42938$, df = 1, $p$-value = 0.5123 (ns), logistic regression). However, the ILS was present significantly more often in the left than the right hemisphere in the chimpanzee ($\chi^2 = 10.419$, df = 1, $p$-value = 0.001247, logistic regression). Finally, in Hominoidea, when a PCGS was present, the ILS was almost always absent (Fig. 2b) (human: presence of ILS in hemispheres with a PCGS versus hemispheres without a PCGS: $\chi^2 = 30.221$, df = 1, $p$-value = 3.855e−08; chimpanzee: $\chi^2 = 14.011$, df = 1, $p$-value = 0.0001817, logistic regression).

**Dorsal medial frontal cortex (MFC).** There are several lines of evidence suggesting that the dorsal MFC is comparable in terms of cytoarchitectonic areal distribution, connectivity, and functional processing between the macaque and human brains[25–28]. The present analysis shows that the sulcal organization of this region is also well conserved.

In the human brain, posterior to the genu of the corpus callosum, we identified four sulci vertical to the CGS or the PCGS when present: the paracentral sulcus (PACS), the pre-paracentral sulcus (PRPACS), the posterior (VPCGS-P), and the anterior (VPCGS-A) vertical paracingulate sulci (Fig. 3). Each one of these sulci was defined as fully present, as a spur or as a dimple when only superficially evident, or absent. For this analysis, we considered the presence of spurs or dimples, which were observed almost exclusively in the baboon and macaque samples, as precursors of sulci that find full expression in the chimpanzee and human brains (see below and Supplementary Fig. 1). Our argument for considering spurs and dimples in macaque and baboon brains as precursors of each one of the four sulci examined comes from an analysis of their normalized spatial location relative to human and chimpanzee brains (see below).

Specifically, we observed that the PACS, PRPACS, VPCGS-P, and VPCGS-A sulci, spurs, or dimples, could be reliably located with respect to four anatomical landmarks: the rostral limit of the pons, the anterior commissure, the caudal limit of the genu of the corpus callosum, and the rostral limit of the corpus callosum (see "Methods" and Fig. 3).

When evaluating the probability of occurrence of each one of the four vertically oriented sulci along the CGS or PCGS, significant species differences were found for PACS (dependent variable: presence of sulci (0/1), main effect species: $\chi^2 = 43.068$, df = 3, $p = 2.381e−09$), PRPACS ($\chi^2 = 49.192$, df = 3, $p = 1.187e−10$), VPCGS-P ($\chi^2 = 108.8$, df = 3, $p = 2.2e−16$), and VPCGS-A ($\chi^2 = 83.311$, df = 3, $p = 2.2e−16$) (see Fig. 3a–d). For PACS, VPCGS-P, and VPCGS-A, post-hoc analysis indicated that these sulci are present in a higher proportion of human brains compared to chimpanzee, macaque, and baboon brains. Additionally, the percentage of chimpanzee brains displaying a PACS, VPCGS-A, and VPCGS-P was significantly higher compared to macaque and baboon brains, but there was no significant difference in the probability of occurrence of these sulci between baboon and macaque brains. For PRPACS, post-hoc analysis indicated that its probability of occurrence was significantly higher in human and chimpanzee brains compared to baboon and macaque brains. There were no significant differences in the occurrence of PRPACS between human and chimpanzee brains, nor between baboon and macaque brains.

In summary, the four vertical sulci, or their precursors, can be found across the four-primate species examined in the current study. In human brains, the probabilities of observing the vertical sulcus PACS versus PRPACS versus VPCGS-P versus VPCGS-A are similar (dependent variable: presence of sulci (0/1), main effect vertical sulci (PACS, PRPACS, VPCGS-P, VPCGS-A): $\chi^2 = 4.6815$, df = 3, $p = 0.1967$). By contrast, these probabilities vary in chimpanzee brains ($\chi^2 = 68.917$, df = 1, $p = 7.279e−15$), baboons

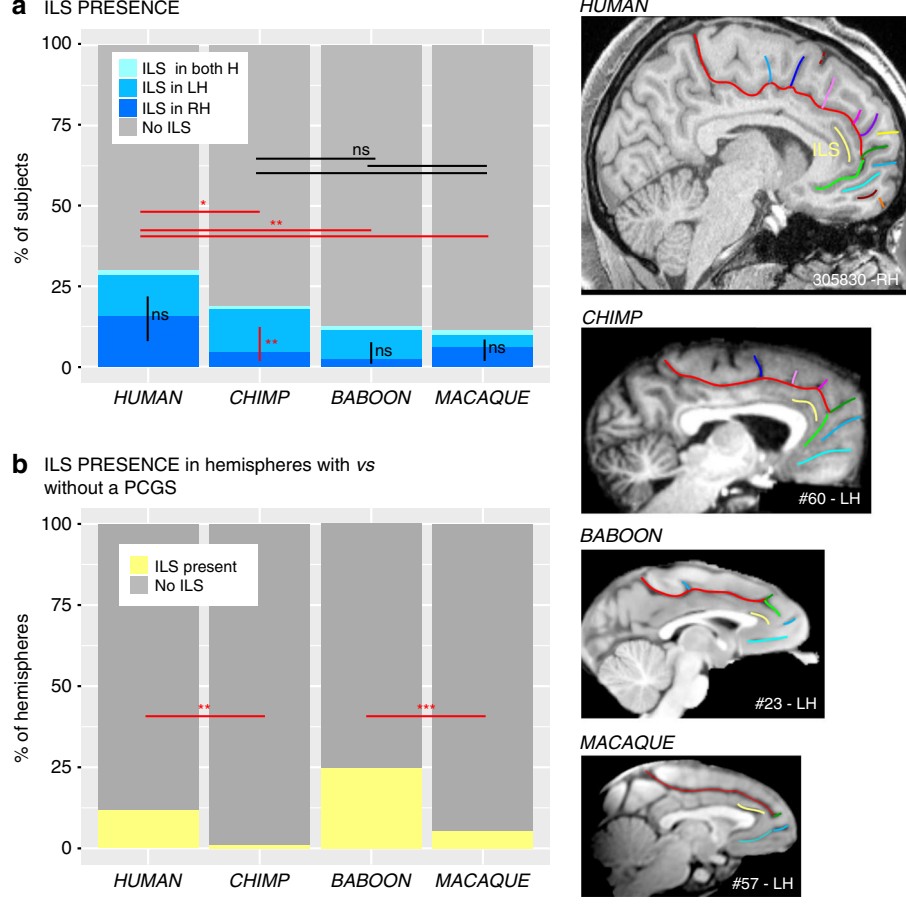

**Fig. 2** Presence of the ILS across primates. The location of the ILS is shown in yellow in typical hemispheres of the human, chimpanzee, baboon, and macaque monkey brains (right panels). **a** The probability of occurrence of an ILS is higher in human compared with non-human primate brains. A post-hoc Tukey test indicated no significant difference in the presence of the ILS in the three non-human primate species. Only in chimpanzee brains, the probability of observing an ILS is higher in the left than in the right hemisphere. No difference between the probability of occurrence of ILS in the left versus right hemisphere was found in human, baboon, and macaque brains. **b** In Hominoidea, when a PCGS is present, the ILS is almost always absent. Statistics: *$p < 0.05$, **$p < 0.01$, ***$p < 0.001$, ns non-significant. Source data are provided as a Source Data file

($\chi^2 = 38.617$, df $= 1$, $p = 2.092e{-}08$), and macaque brains ($\chi^2 = 67.617$, df $= 1$, $p = 1.382e{-}14$). As can be seen in Fig. 4a–d, PACS, PRPACS, and VPCGS-A are the most conserved sulci across primate species, while VPCGS-P is the least conserved sulcus. Chimpanzee brains display a sulcal organization of the dorsal MFC comparable to that of the human brain, and macaque and baboon brains display evidence of the emergence of this organization. Importantly, vertical sulci emerging from the CGS or the PCGS located in the dorsal MFC can be identified in relation to fixed anatomical landmarks that are consistent across primates.

**Ventral mPFC**. The anterior cingulate cortex (ACC), the ventromedial prefrontal cortex (vmPFC), and the medial frontopolar cortex (mFPC) are the focus of many studies on, respectively, emotional processing[29], value-based decision-making[30,31], and high-order socio-cognitive processing (mentalizing)[23,32,33]. These processes reach their epitome in human subjects, and the present analysis of sulcal organization provides insights into the evolution of the above mentioned cortical regions and thus the anatomical substrate for these higher order processes in the primate order.

Anterior to the genu of the corpus callosum, we identified in human brains up to 9 distinct sulci[34]. One of the distinct features of the vmPFC is the branching of sulci at the end of the CGS. In human brains, the rostral end of the CGS is characterized by two

sulci forming a downward facing bifurcation: the supra-rostral sulcus (SU-ROS) and the supra-orbital sulcus (SOS) (Fig. 5a). When the PCGS is absent, the fork is located at the rostral end of the CGS but, when the PCGS is present, it is more frequently located at the rostral end of the PCGS (Supplementary Fig. 2).

In non-human primates, the majority of the brains displayed different sulcal patterns at the rostral end of the CGS. The form and orientation of these folds were highly variable between species. Because of this discrepancy with the observed pattern in human brains, we chose to label the sulci at the end of the CGS, the ventral (CGS-VE) and dorsal (CGS-DE) extensions of the cingulate sulcus. We viewed the CGS-VE and the CGS-DE in the nonhuman primate brains as precursors of the human SU-ROS and SOS, respectively.

In chimpanzee, the fork formed by the CGS-VE and CGS-DE extensions of the cingulate sulcus was also oriented downwards in the majority of hemispheres (81.25%). However, in 17.5% of hemispheres, the fork was forward facing, and was incomplete (i.e., CGS-DE is present but CGS-VE is absent) in 1.25% of hemispheres (Fig. 5a). Interestingly, as in human brains, when the PCGS is absent, the fork is located at the rostral end of the CGS. By contrast, when the PCGS is present in chimpanzee, the fork is still located at the rostral end of the CGS—and not at the PCGS as in human—brains in the majority of hemispheres (Supplementary Fig. 2).

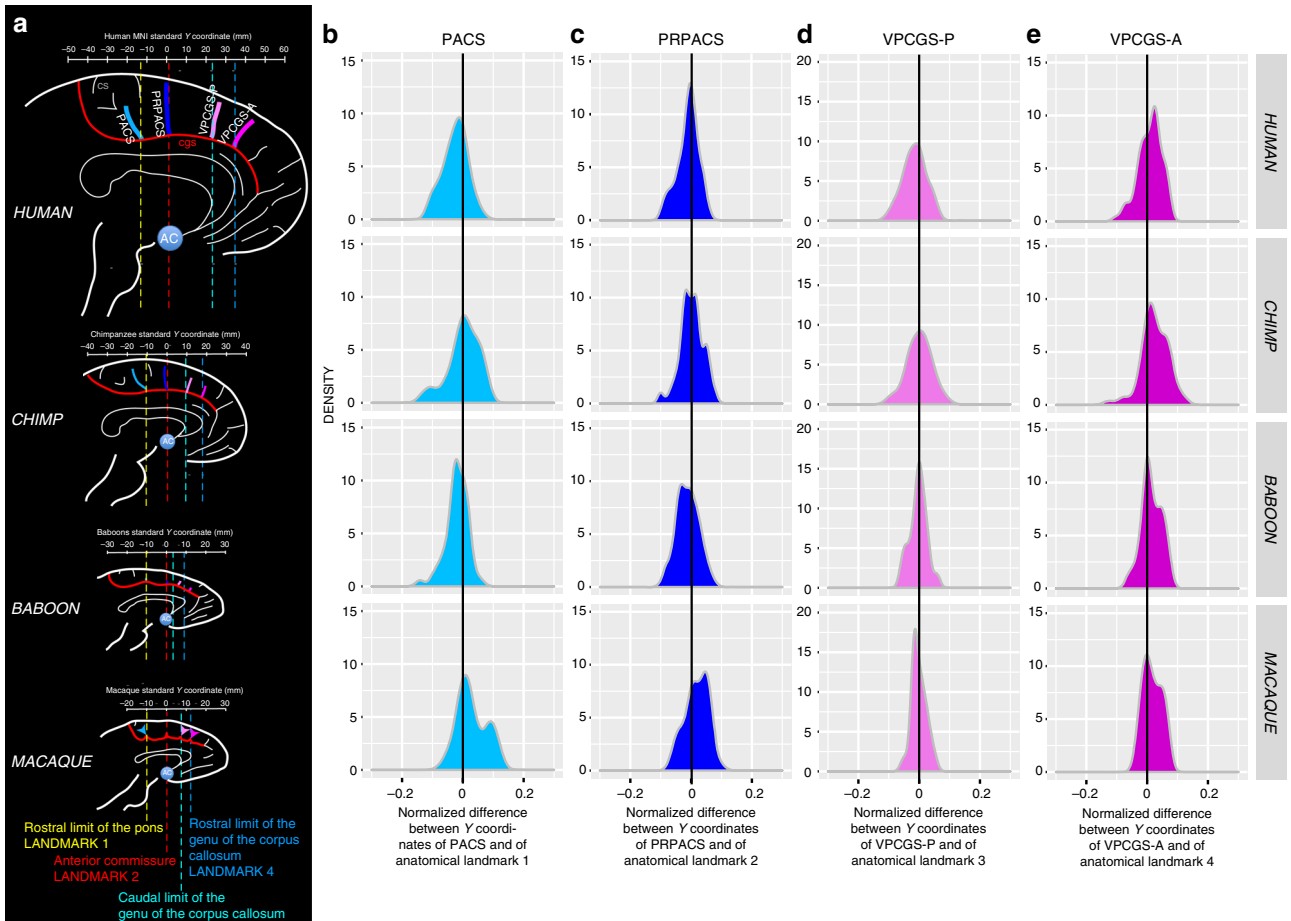

**Fig. 3** Location of vertical sulci in the dorsal MFC across primates. **a** Schematic representation of the location of the various sulci in the respective standard space of each primate species. The scale represents the antero-posterior level (in mm) in each brain. Fixed anatomical landmarks across primates are displayed: the rostral limit of the pons (landmark 1, in yellow), the anterior commissure (landmark 2, in red), the caudal limit of the genu of the corpus callosum (landmark 3, in light blue), and the rostral limit of the genu of the corpus callosum (landmark 4, in dark blue). **b** Density of the difference between the Y coordinates of PACS and of the anatomical landmark 1 in human, chimpanzee, baboon, and macaque (from top to bottom panels). This difference was normalized in relation to the brain size (total antero-posterior extent). The 0 value corresponds to the Y level of the anatomical landmark 1 in standard brains. **c** Density of the normalized difference between the Y coordinates of PRPACS and of the anatomical landmark 2. The 0 value corresponds to the Y level of the anatomical landmark 2 in standard brains. **d** Density of the normalized difference between the Y coordinates of VPCGS-P and of the anatomical landmark 3. The 0 value corresponds to the Y level of the anatomical landmark 3 in standard brains. **e** Density of the normalized difference between the Y coordinates of VPCGS-A and of the anatomical landmark 4. The 0 value corresponds to the Y level of the anatomical landmark 4 in standard brains. Source data are provided as a Source Data file

In the baboon (*P. papio*), the fork formed by CGS-VE and CGS-DE is also oriented downwards in the majority of hemispheres but to a lesser extent (40%) compared to the human and chimpanzee brains. In 25% of hemispheres, the fork is forward facing, incomplete in 28.75% of hemispheres, and absent in 6.25% of hemispheres (Fig. 5a). In the rhesus monkey (*M. mulatta*), the fork formed by CGS-VE and CGS-DE displays the same patterns as in the baboon, but in different proportions: the most frequent pattern observed is a forward-facing fork (in 45% of hemispheres), followed by a fork oriented downwards (in 23.75% of hemispheres). The fork is incomplete in 13.75% of hemispheres and is absent in 17.5% of hemispheres (Fig. 5a).

The assessment of the relative location of the intersection of this fork with the CGS or PCGS across primates (see "Methods"), revealed that it is located at the level of the rostral limit of the genu of the corpus callosum in humans (Fig. 5b), but is located dorsal to it in the non-human primate brains ($F_{(3,298)} = 43.506$, $p < 2.2e−16$, GLM with species as fixed effect and the difference

between the Z value where this intersection is found and the Z value where the rostral limit of the genu of the corpus callosum is found as dependent variable), strongly suggesting a downward migration in the human brain.

Ventral to SU-ROS or CGS-VE lies the superior rostral sulcus (ROS-S). The ROS-S is a long and deep sulcus that is present in nearly 100% of hemispheres across all four species and, therefore, does not differ significantly in its occurrence (main effect of species: $\chi^2 = 4.5334$, df = 3, $p = 0.2093$) (Fig. 6a). ROS-S is often connected to the accessory supra-orbital sulcus (ASOS). The ASOS is equally present in human (88.75%) and chimpanzee (81.25%) hemispheres and is observed significantly less often in baboon (26.25%) and macaque (21.25%) hemispheres compared to chimpanzees and humans (main effect of species: $\chi^2 = 132.81$, df = 3, $p$-value = 2.2e−16, logistic regression) (Fig. 6b). Ventral to ROS-S, the inferior rostral sulcus (ROS-I) is equally present in human (35%) and chimpanzee (37.5%) hemispheres, but largely absent in baboon (3%) and macaque (1.5%) ($\chi^2 = 60.348$, df = 3, $p = 4.952e−13$) (Fig. 6c).

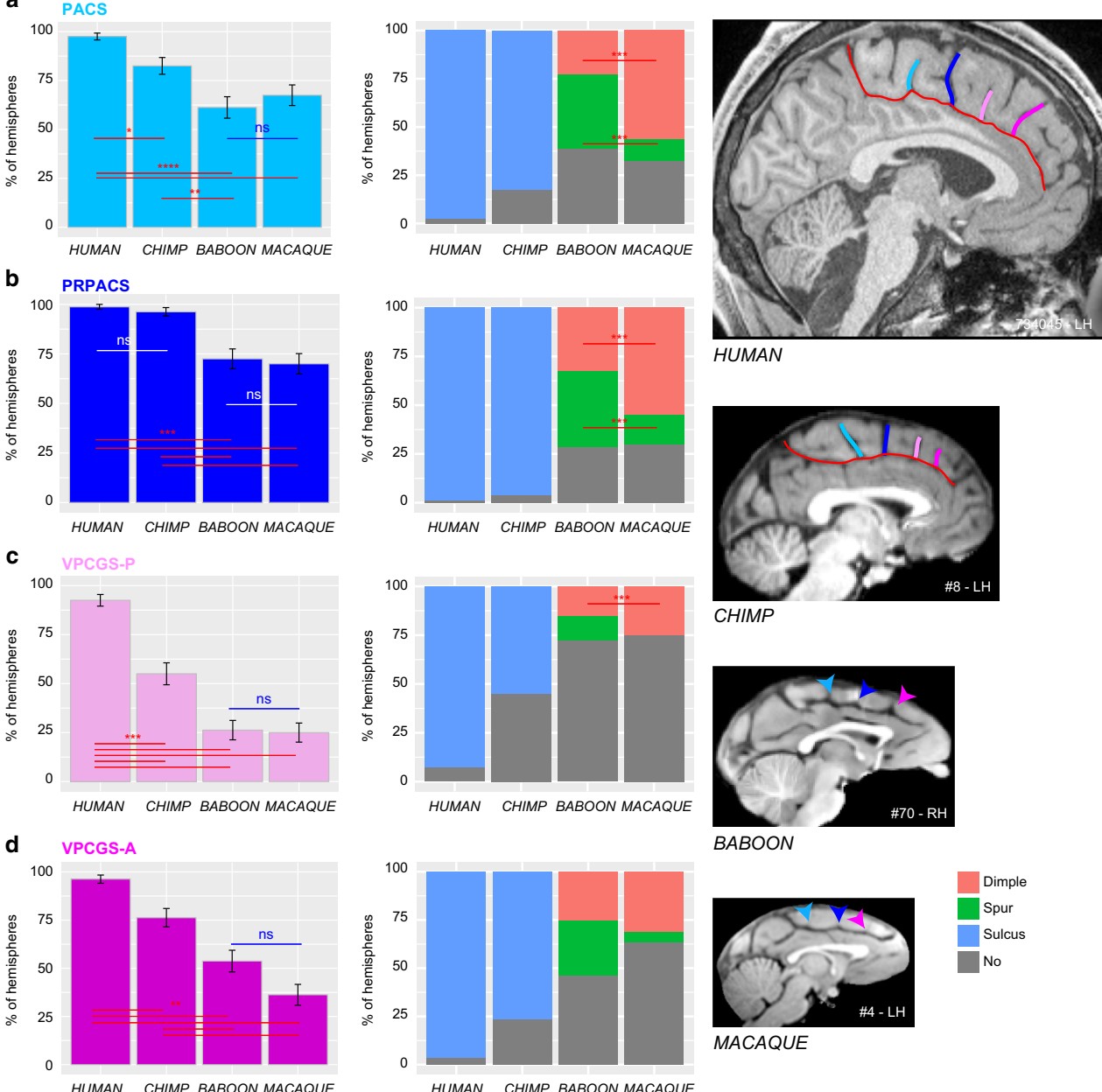

**Fig. 4** Presence of vertical sulci in the dorsomedial frontal cortex across primates. Probability of occurrence (±s.e.m.) of the paracentral sulcus (PACS, **a**), the pre-paracentral sulcus (PRPACS, **b**), the posterior (VPCGS-P, **c**) and the anterior (VPCGS-A, **d**) vertical paracingulate sulci in the four species. **a** PACS. Left panel: The probability of observing a PACS decreases from human, chimpanzee, baboon, to macaque. Posthoc analysis showed that the presence of PACS was higher in Hominoidea (human and chimpanzee) than in Old-world monkeys (baboon and macaque), but that it was similar between human and chimpanzees and also between baboon and macaque. Right panel: PACS characteristics vary across primates: it is a sulcus (blue) in 100% of hemispheres in Hominoidea, it is a spur (green) or a dimple (pink) in Old-world monkeys. It is more frequently a spur than a dimple in baboon, as opposed to macaque. **b** PRPACS. Left panel: the probability of observing PACS is higher in Hominoidea than in Old-world monkeys but is similar between human and chimpanzee, on one hand, and between baboon and macaque on the other hand. Right panel: PRPACS characteristics vary across primates: it is a sulcus in 100% of hemispheres in Hominoidea, but it is a spur or a dimple in Old-world monkeys. It is more frequently a spur than a dimple in baboon, as opposed to macaque. **c** VPCGS-P. Left panel: the probability of observing VPCGS-P decreases from human, chimpanzee, baboon, to macaque. Posthoc analysis demonstrated that the presence of the VPCGS-P was higher in human than in non-human primates. Its probability of occurrence was also higher in chimpanzee than in Old-world monkeys. It was, however, similar between macaque and baboon. Right panel: VPCGS-P characteristics vary across primates: it is a sulcus in 100% of hemispheres in Hominoidea, but it is a spur or a dimple in baboon and only a dimple in Macaque. **d** VPCGS-A. Left panel: The probability of occurrence of VPCGS-A decreases from human, chimpanzee, baboon, to macaque. Right panel: VPCGS-P characteristics vary across primates: it is a sulcus in 100% of hemispheres in Hominoidea, but it is more frequently a spur than a dimple in baboon, as compared with macaque. Statistics: *$p < 0.05$, **$p < 0.01$, ***$p < 0.001$, ns non-significant. Source data are provided as a Source Data file

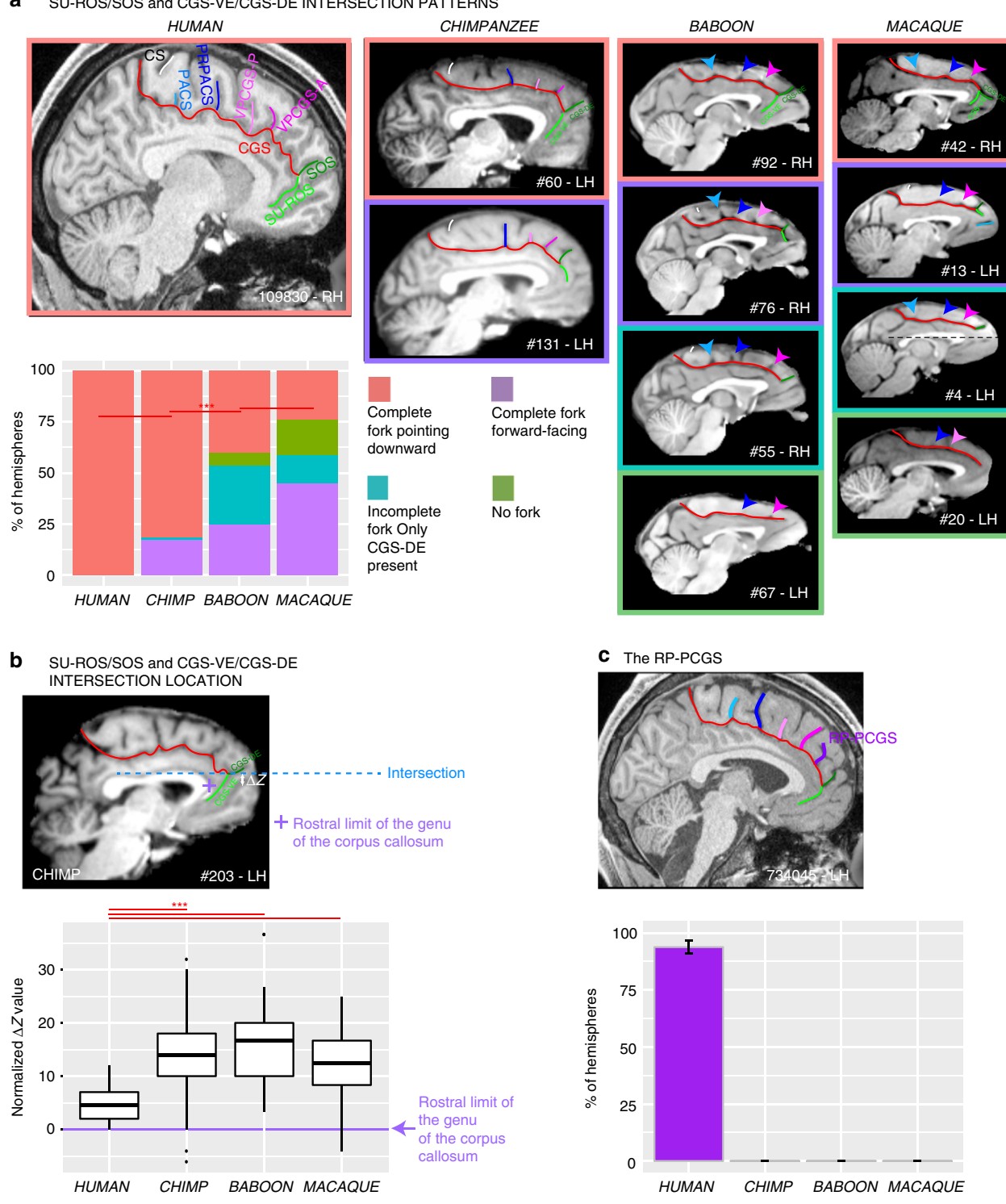

**a** SU-ROS/SOS and CGS-VE/CGS-DE INTERSECTION PATTERNS

HUMAN · CHIMPANZEE · BABOON · MACAQUE

**b** SU-ROS/SOS and CGS-VE/CGS-DE INTERSECTION LOCATION

**c** The RP-PCGS

Finally, we observed at the most rostral part of the frontal cortex, several sulci that are present almost exclusively in Hominoidea. Notably, we observed, exclusively in the human brain, an additional vertical sulcus joining the CGS or PCGS anterior to the rostral limit of the genu of the corpus callosum which was labeled the rostro-perpendicular paracingulate sulcus (RP-PCGS) (Fig. 5c). In addition, we observed a sulcus located just dorsal to SOS, which joins the lateral surface of the brain but not the CGS or PCGS, called the dorsomedial polar sulcus (DMPS) in 88.75% of hemispheres in human and in 48.75% of

hemispheres in chimpanzee. By contrast, only 2.5% of the hemispheres of baboon brains display this sulcus and no hemispheres in macaque brains (main effect of species: $\chi^2 = 227.43$, df = 3, $p$-value = 2.2e−16). A second sulcus, the ventromedial polar sulcus (VMPS), located in the ventral MFC, joining the lateral surface of the brain and sometimes the ROS-I, was identified more frequently in human brains (57.5% of hemispheres) than in chimpanzee brains (11.25% of hemispheres) and was absent in the baboon and the macaque (main effect of species: $\chi^2 = 126.29$, df = 3, $p = 2.2e−16$) (Fig. 7c). Figure 8

**Fig. 5** Morphological characteristics of the junction between the dorsal and ventral MFC. **a** Rostral end of the CGS/PCGS. In human. The rostral end of CGS is characterized by two sulci forming a fork pointing downward: SU-ROS, pointing downward, and SOS, pointing upward. In chimpanzee. The fork formed by the precursors of SU-ROS and SOS, i.e., CGS-VE and CGS-DE, respectively, is also pointing as in the human brain in the majority of hemispheres. In 17.5% of hemispheres, the fork is forward facing (purple area), and is incomplete (green area) (i.e., SOS is present but SU-ROS is absent) in 1.25% of hemispheres. In baboon. The fork formed by CGS-VE and CGS-DE is also pointing downward in the majority of hemispheres but to a lesser extent than in Hominoidea. In 25% of hemispheres, the fork is forward facing, it is incomplete in 28.75% of hemispheres, and is absent in 6.25% of hemispheres. In macaque. The fork formed by CGS-VE and CGS-DE is displaying the same patterns as in the baboon but in different proportions: the most frequent pattern observed is a fork facing forward, followed by a fork pointing downward. The fork is incomplete in 13.75% of hemispheres and is absent in 17.5% of hemispheres. Statistics: $*p < 0.05$, $**p < 0.01$, $***p < 0.001$, ns non-significant. **b** Location of the SU-ROS/SOS and CGS-DE/CGS-VE intersection. The top diagram shows the location of the intersection between SU-ROS/SOS and CGS-VE/CGS-DE in a typical hemisphere of a chimpanzee. The $\Delta Z$ corresponds to the difference between the dorsoventral $Z$ coordinates where the intersection is observed and the dorsoventral $Z$ coordinates where the rostral limit of the genu of the corpus callosum (represented by a purple cross) is located. $\Delta Z$ was then normalized for brain size to compare across primate brains. Boxplots displaying the mean (±s.e.m.) normalized $\Delta Z$ across individuals are presented in the bottom diagram. Note that black dots represent outliers and horizontal lines the mean of each distribution. **c** Probability of occurrence (±s.e.m.) of RP-PCGS. The RP-PCGS is human-specific since it is not present in non-human primates. Statistics: $*p < 0.05$, $**p < 0.01$, $***p < 0.001$, ns non-significant. Source data are provided as a Source Data file

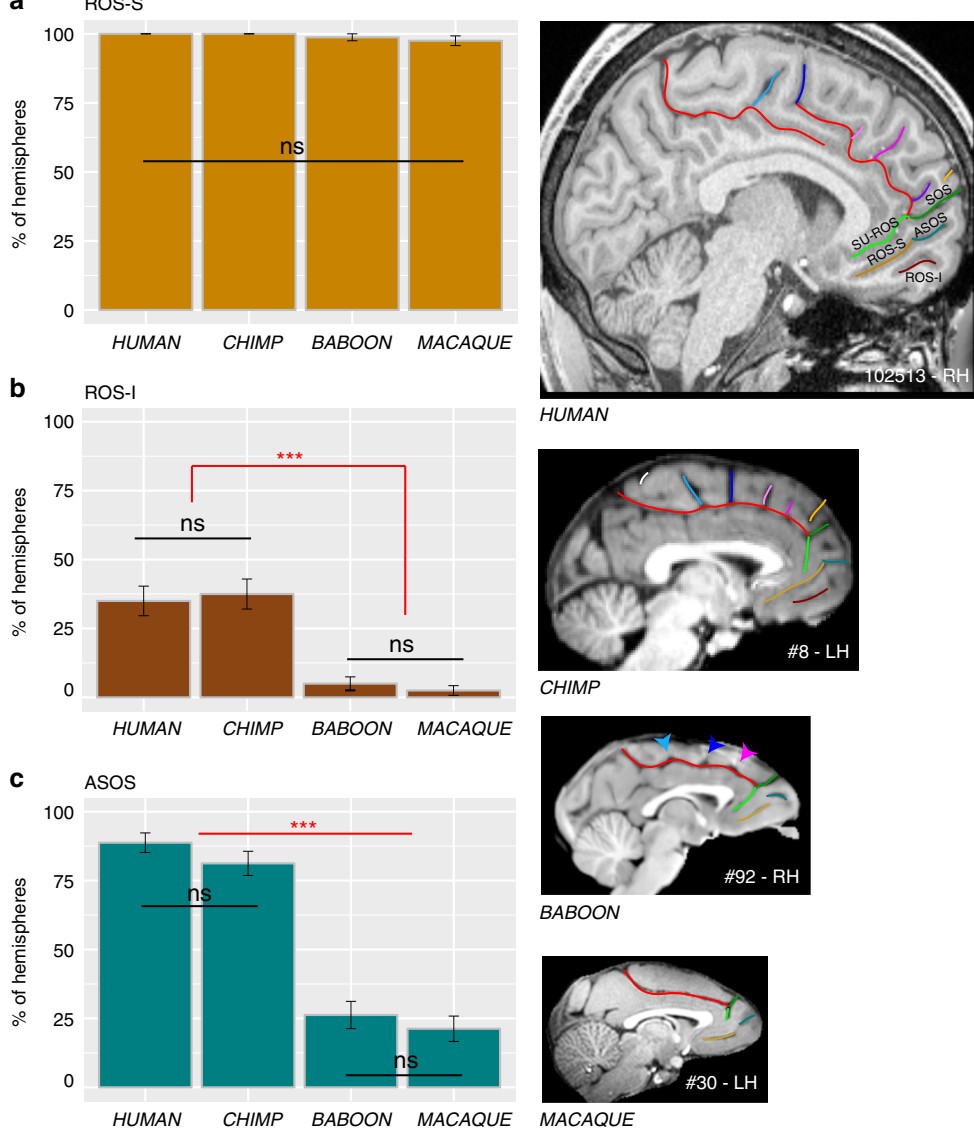

**Fig. 6** Sulci present in the ventral MFC across primates. The location of the ROS-S, ROS-I, and ASOS sulci are displayed in human, chimpanzee, baboon, and macaque brains in the right panels. **a** The probability of occurrence (±s.e.m.) of ROS-S is similar in human and non-human primates. **b** The probability of occurrence (±s.e.m.) of ROS-I is comparable in human and chimpanzee and is almost absent in Old-world monkeys. **c** The probability of occurrence of ASOS is equal in human and chimpanzee. It is decreased in Old-world monkeys but equally present in baboon and macaque. Statistics: $*p < 0.05$, $**p < 0.01$, $***p < 0.001$, ns non-significant. Source data are provided as a Source Data file

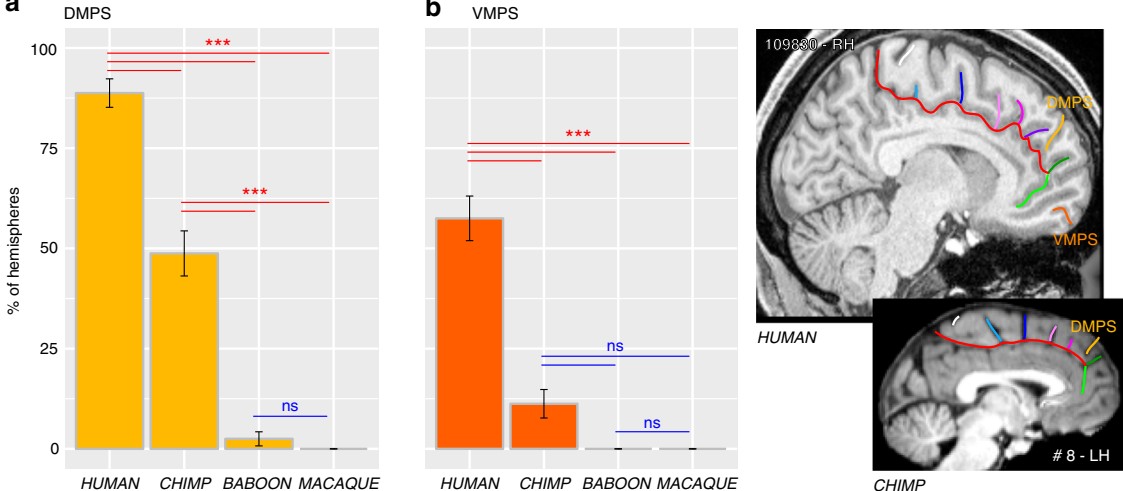

**Fig. 7** Probability of occurrence of DMPS and VMPS across primates. The location of the DMPS and VMPS sulci are displayed in human and chimpanzee brains in the right panels. The probability of occurrence of both DMPS (**a**) and VMPS (**b**) is higher in human than in chimpanzee but is very low in Old-world monkeys. Statistics: *$p < 0.05$, **$p < 0.01$, ***$p < 0.001$, ns non-significant. Source data are provided as a Source Data file

summarizes the major changes occurring between macaque, baboon, chimpanzee and human brains.

## Discussion

Analysis of sulcal brain morphology has proven to be a powerful method for a better understanding of the evolution of the MFC in primates. Although gyrification correlates with brain volume[1,35], and therefore with the expansion of primate neocortex, the present results did not reveal random or homogeneous changes in sulcal morphology in the MFC across the four primate species examined (e.g., differences in the ACC/vmPFC but no differences in the MCC/medial premotor cortex). The observed differences were in specific regions that have expanded more during evolution[36]. Here, we report that chimpanzees (*P. troglodytes*) possess a PCGS, a feature previously thought to be unique to the human brain (*H. sapiens*)[26–28]. Two differences must, however, be emphasized: (1) the probability of occurrence of a PCGS is lower in the chimpanzee than in the human brain, and (2) the presence of a PCGS is largely lateralized in the left hemisphere in the human brain, whereas it is equally present in both hemispheres in the chimpanzee (Fig. 1e). The leftward asymmetry observed in the human brain appears to be a human trait under the influence of genetic factors and the in-womb environment[14]. Although its functional significance is not fully understood, it has been shown that it correlates with the involvement of the left cingulate cortex in language tasks in right-handed subjects[37]. Notably, only in the chimpanzee brain, the probability of occurrence of an ILS is higher in the left than in the right hemisphere (Fig. 2a). Since the ILS is observed only in hemispheres displaying no PCGS in human and chimpanzee brains (Fig. 2b), one can hypothesize that the lateralized presence of an ILS in the left hemisphere in chimpanzee brains may constitute an evolutionary reflection of the beginnings of the sulcal organization observed in the human brain, i.e., a PCGS lateralized in the left hemisphere and an ILS not lateralized.

Another major difference between the primate species examined was the organization of the MFC rostral to the genu of the corpus callosum. Differences were associated with (1) a sulcus only observed in the human brain, the RP-PCGS, (2) the progressive emergence of the DMPS and VMPS, and (3) the SU-ROS/SOS intersection that is displaced downwards from the level of the rostral limit of the genu of the corpus callosum in humans

in comparison with more distantly related primates. There is no evidence from cytoarchitectonic[38,39] and neuroimaging[24,40] studies that these changes are associated with new cortical areas, suggesting instead differential expansion of (1) the rostral MFC (area 9) and mFPC (area 10), two brain regions that have been considered important for high-order socio-cognitive processing[23,32,33] and (2) the vmPFC implicated in value-based decisions[30,31]. Our results are in line with a recent study showing that the ACC/vmPFC is a region of high structural variability within the human brain and the macaque brain[41]. The present study provides an explanation of the nature of this variability: Since we observed reduced variability in the ACC/vmPFC of human and chimpanzee brains compared to Old-world monkeys, we suggest that a facing downward SU-ROS/CGS-VE morphology might be associated with a trait that is conferring an evolutionary advantage, and was therefore selected.

Furthermore, the present data demonstrate that, in Old-world monkeys, in contrast to the ACC/vmPFC region, the organization of the MFC posterior to the genu of the corpus callosum (which corresponds to the MCC, where the medial motor and premotor areas lie), has all the basic features of the human sulcal organization and reaches a similar organization across Hominoidea. Specifically, Old-world monkeys display precursors—in the form of spurs or dimples—of the four vertical sulci of this region: PACS, PRPACS, VPCGS-P, and VPCGS-A. This result provides additional evidence that the MCC is comparable both anatomically and functionally across macaque and human brains (see Procyk et al.[25] for a detailed anatomo-functional comparison), contrary to what had been proposed before[26–28]. Importantly, we show that these sulci refer to specific and highly-reliable anatomical features, and we provide a simple framework within which they can be identified: PACS is located at the level of the rostral limit of the pons, PRPACS at the level of the anterior commissure, VPCGS-P and VPCGS-A at the level of the caudal and rostral limits of the genu of the corpus callosum, respectively. These landmarks could be used to guide interpretation of functional neuroimaging data and, also, comparative functional studies across primates.

The present results also show that the ventral MFC is highly conserved from macaque to human brains. ROS-S, which is in the center of this region, is a deep sulcus in all primate species. Our results are supported by recent studies showing that the main vmPFC node of the default mode network lies in this sulcus in the

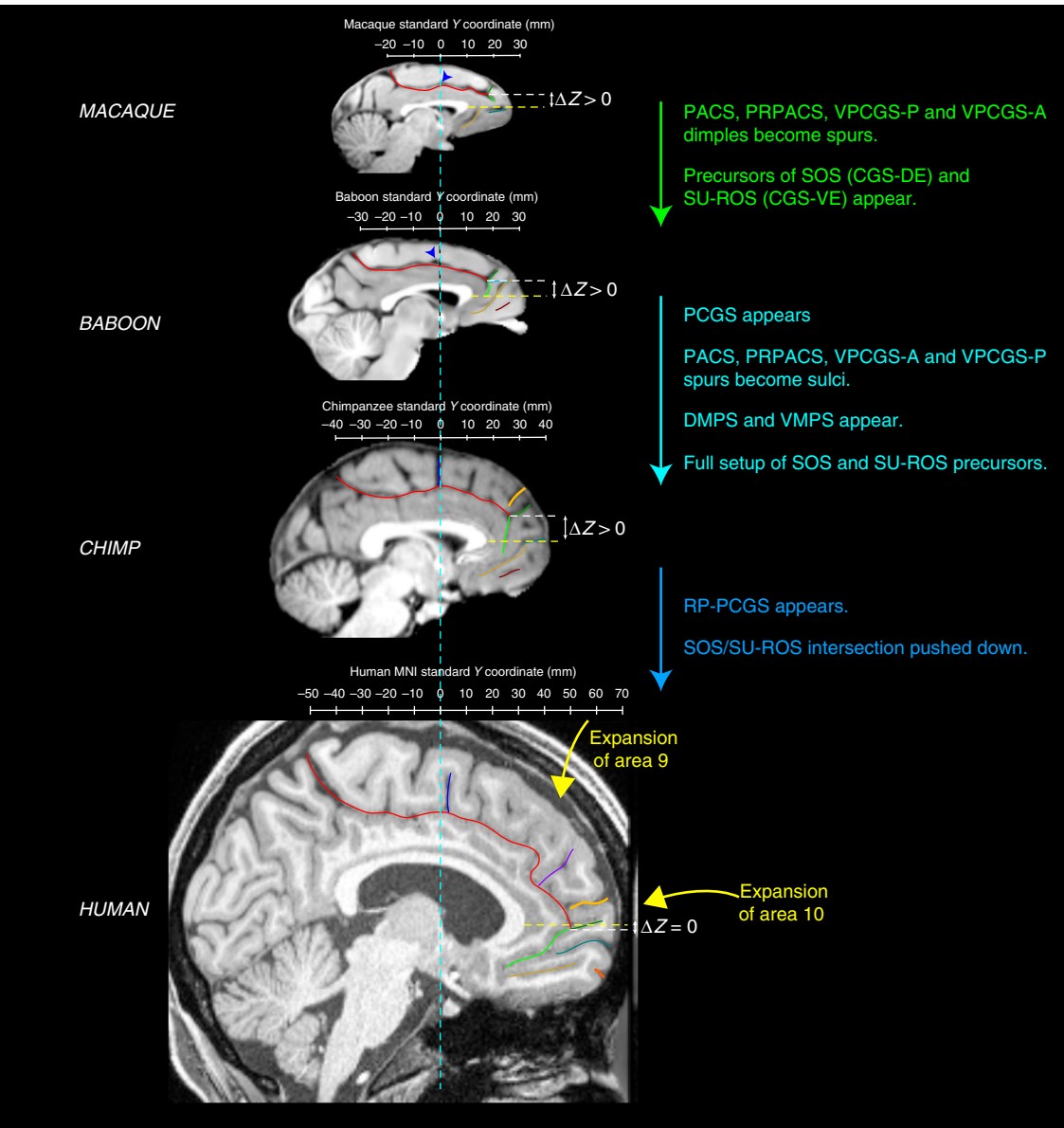

**Fig. 8** Summary of sulcal changes occurring from macaque to human. Schematic sulcal organization of the MFC in each species is represented in their respective stereotaxic space. Main changes from one species to another are indicated. The $\Delta Z$ corresponds to the difference between the dorso-ventral $Z$ coordinates where the intersection is observed and the dorso-ventral $Z$ coordinates where the rostral limit of the genu of the corpus callosum is located. The difference $\Delta Z$ is decreased in the human compared to non-human primates, showing that the intersection between the CGS with the SU-ROS/CGS-VE and with the supra-orbital sulcus SOS/CGS-DE is pushed down from non-human to human primates

human brain[34] and that the macaque default brain network is comparable to that observed in the human brain[42,43].

Collectively, the results presented here provide (1) strong evidence for a comparable organization of the MCC across primates and (2) a clear framework to test hypotheses regarding the relation between the vertical sulci/spurs/dimples, cytoarchitectonic areas, and functions. For example, there is some evidence that PACS may be a sulcus occurring at the border between medial area 4 and medial area 6 in several primate species: human[44-46], orangutan[45,47], and macaque[48]. Finally, as no new areas have been identified in the human MFC in neuroimaging[24,40] and cytoarchitectonic [38,39] studies, the emergence of a new sulcus in the human MFC might reflect the relative expansion of this region compared to other primates. For instance, we suggest that

the RP-PCGS reflects the expansion of medial area 9, a region thought to be important for social and metacognitive processes[23,32,33].

Future studies may test for relationships between the vertical sulci in the dorsal MFC and cytoarchitectonic areas. Based on cytoarchitecture, Vogt has proposed a hierarchical four-region organization of the cingulate cortex[28] both in human and macaque brains emphasizing the distinction between the ACC, the anterior and the posterior MCC (aMCC and pMCC), the posterior cingulate cortex (PCC), and the retrosplenial cortex. We superimposed this model on the present sulcal organization scheme in the human and macaque brains to examine whether the vertical sulci may have significant meaning in terms of the anatomo-functional organization of the cingulate cortex. Based

on the consistent location of the vertical sulci in the dorsal MFC, one can hypothesize that, in both the human and macaque brains, (1) the VPCGS-A, which is located at the rostral limit of the genu of the corpus callosum and is better conserved than the VPCGS-P across primates, may be a reliable landmark to distinguish the ACC from the aMCC; (2) the PRPACS, which is located at the level of the anterior commissure and highly conserved across primates, may correspond to the limit between aMCC and pMCC; (3) the PACS, which is located at the level of the rostral limit of the pons, might be the limit between pMCC and PCC. Although there is no currently available information about the four-region model in the MFC of the baboon (*P. papio*) and chimpanzee (*P. troglodytes*), one can infer, based on our demonstration that the vertical sulci examined are conserved and hold the same location as in the human and macaque brains, that these sulci may limit the same anatomo-functional regions. This hypothesis is displayed in Supplementary Fig. 3 and future studies can test directly this hypothesis. It is important to note, however, that the precise relation of sulci to cytoarchitectonic areas remains controversial[49] and requires considerably more investigation in appropriately sectioned brains, i.e., in histological sections perpendicular to the orientation of a given sulcus, as originally pointed out by Economo and Koskinas[50].

Besides sulci, we propose that other anatomical landmarks could help guide interpretation of the functional organization of the MFC. This is a relevant question as New-World monkeys include species with gyrified brains (e.g., Capuchin and Squirrel monkeys) and species with lissencephalic brains (e.g., the marmoset)[51,52]. Importantly, it has been shown that the relative position and cytoarchitecture of homologous areas in New versus Old-world monkeys are similar[53,54], despite a different brain size[51]. On the basis of the framework provided by the present study, it would be reasonable to hypothesize that homologous cytoarchitectonic and functional regions in the MFC in New World monkeys may be identified using landmarks common to all primates: the PCC/MCC limit might be located at the level of the rostral part of the pons (where PACS is found in Old-world monkeys, apes, and human). The MCC should expand from this location to the anterior part of the genu of the corpus callosum, i.e., where the VPCGS-A is found and which corresponds to the limit between MCC and ACC. As in humans, we also predict the border between supplementary motor area and pre-supplementary motor area to be at the level of the anterior commissure in standardized stereotaxic space, i.e., at $y = 0$ in Talairach and Tournoux coordinates[55] or $y = +3$ in the latest asymmetrical MNI brain[56].

Manual labeling of the sulci following visual inspection in the various species in the present study may appear as a limitation. Yet, a similar strategy has been recently employed and confirmed the organization of these sulci in the human brain[34]. Furthermore, such precise analysis of sulci in multiple species represents an important step forward towards the development of efficient algorithms for automatic labeling, which are currently not available. To characterize further the evolution of sulcal patterns in primates, future studies will need to consider the assessment of more species. But, as no endocast exists for the MFC, it is important to note that only analysis on living species could be conducted to understand how structures on the medial surface of the brain have evolved.

Altogether comparative neuroanatomy studies show a relatively well conserved anatomo-functional organization of the MFC in primates[24,38,40], which suggests that the precursor of the human brain was already present in the last ancestor to humans, apes, and Old-world monkeys. Differences in sulcal pattern morphology as observed in the present study could be related to the expansion of cortical areas, which in turn could be related to

expansion in motor and cognitive processes. Importantly, the brain size difference across primate species does not explain by itself the pattern of sulcal organization observed in the present study as we observed (1) different patterns of sulcal morphology within each species (controlling therefore for brain size), and (2) a consistent antero-posterior organization of the vertical sulci and their precursors in the dorsal MFC in the four species (i.e., in different brains of different sizes). Studies assessing within species inter-individual variability in sulcal morphology and behavioral performance provide evidence of the existence of relationships between structure and function. Such evidence exists for the human brain, e.g., refs. [57,58]. and the chimpanzee brain, e.g., ref. [59]. Indeed, Hopkins et al.[59] found that male chimpanzees producing attention getting sounds showed a larger leftward asymmetry in the depth of central and ventral portions of the central sulcus compared to males that produce less attention getting sounds with females showing the opposite pattern. We also recently showed that the presence of a PCGS and an intra-limbic sulcus, particularly in the left hemisphere, was associated respectively with the production and use of attention-getting sounds by chimpanzees as well as rightward handedness (Hopkins et al., in preparation). It is therefore reasonable to think that interindividual variability in sulcal morphology in macaque and baboon may be associated with variability in behavioral performance.

The present study provides critical new evidence in the context of a new comprehensive framework regarding prefrontal cortical evolution. On one hand, these results are in agreement with previous comparative connectomic studies[24,40] and show a conserved organization of the MFC, suggesting the existence of a precursor architecture in the last common ancestor to human, apes, and Old-world monkeys. On the other hand, the appearance in human and ape brains of the PCGS, and the relative expansion of the rostral and dorsal part of the MFC in the human brain (emergence of the RP-PCGS and the more ventral position of the intersection between SUROS/CGS-VE, SOS/CGS-DE and CGS) (summarized in Fig. 8) might reflect the evolution of higher-order mentalizing processes in Hominoidea[60,61] from a rudimentary form in Old-world monkeys[62,63].

To conclude, the present article, by throwing light on how the MFC has evolved, paves the way to future studies assessing relationships between sulcal morphology, behavioral performance, and functional organization within and across primate species. We anticipate that this framework will be used to examine the relationships between sulcal morphology, cytoarchitectonic areal distribution, connectivity, and function, within the studied species and also across the primate order.

## Methods

Neuroimaging T1 anatomical data of 197 human (*H. sapiens*), 225 chimpanzee (*P. troglodytes*), 88 baboon (*P. papio*), and 80 rhesus monkeys (*M. mulatta*) brains were analyzed.

**Human subjects**. High-resolution anatomical scans of the human brain were obtained from the Human Connectome Project database [http://www.humanconnectome.org/]. Only data from subjects displaying no family relationships were analyzed. The participants in the HCP study were recruited from the Missouri Family and Twin Registry that includes individuals born in Missouri[64]. Acquisition parameters of T1 anatomical scans are the following: whole head, 0.7 mm³ isotropic resolution, TR = 2.4 s, TE = 2.14 ms, flip angle = 8° (more details can be found at [https://humanconnectome.org/storage/app/media/documentation/s1200/HCP_S1200_Release_Appendix_I.pdf]). The full set of inclusion and exclusion criteria is detailed elsewhere[64]. Briefly, the HCP subjects are healthy individuals free from major psychiatric or neurological illnesses. They are drawn from ongoing longitudinal studies[64], where they had received extensive assessments, including the history of drug use, and emotional and behavioral problems. The experiments were performed in accordance with relevant guidelines and regulations and all experimental protocols were approved by the Institutional Review Board (IRB) (IRB # 201204036; Title: "Mapping the Human Connectome:

Structure, Function, and Heritability"). All subjects provided written informed consent on forms approved by the Institutional Review Board of Washington University in St. Louis. In addition, the present study received approval (no. 15-213) from the Ethics Committee of Inserm (IORG0003254, FWA00005831) and from the Institutional Review Board (IRB00003888) of the French Institute of Medical Research and Health.

**Non-human primates**. High-resolution anatomical scans of chimpanzee (*P. tro-glodytes*) and baboon (*P. papio*) brains were obtained from the laboratories of Dr. William Hopkins and Dr. Adrien Meguerditchian, respectively. High-resolution anatomical scans of Macaque (*M. mulatta*) brains were obtained from the laboratories of Drs E. Procyk and C. Amiez, W. Hopkins, J. Sallet, F. Hadj-Bou-ziane, and S. Ben Hamed. These data are now available in the PRIMatE Data Exchange (PRIME-DE) database [http://fcon_1000.projects.nitrc.org/indi/indi-PRIME.html][65]. Note that no new data were collected specifically for the purpose of the present study. Data collected initially for studies on rhesus monkeys (*M. Mulatta*) and baboons (*P. papio*) were conducted under local ethics agreements (licenses from the United Kingdom (UK) Home Office; Provence and Lyon ethics committees) and in accordance with The Animals (Scientific Procedures) Act 1986 and with the European Union guidelines (EU Directive 2010/63/EU). Chimpanzee (*P. troglodytes*) data collection was approved by the Institutional Animal Care and Use Committees at YNPRC and UTMDACC and also followed the guidelines of the Institute of Medicine on the use of chimpanzee in research.

**Neuroimaging data analysis**. Human and macaque brains were normalized in the Human (http://www.bic.mni.mcgill.ca/ServicesAtlases/HomePage) and Macaque[66] MNI stereotaxic coordinate systems, respectively. Chimpanzee brains were nor-malized in the Chimpanzee standard brain developed by Dr. W. Hopkins[67] [http://www.chimpanzeebrain.org/]. Baboon brains were normalized in the Baboon standard brain developed by Dr. A. Meguerditchian[68] [http://www.nitrc.org/projects/haiko89/]. Note that normalization of all primate brains consisted in linear registrations, which has the great advantage of allowing within-species compari-sons between brains without altering relationships between sulci and gyri. It is, therefore, unlikely that such processing influences commonality and divergence of sulcal organization observed between species. Normalization of primate brains was performed with SPM12 [https://www.fil.ion.ucl.ac.uk/spm/software/spm12/].

Two levels of analysis were performed:

(1) **A large qualitative analysis** in the whole set of data: 197 human, 225 chimpanzee, 88 baboon, and 80 rhesus monkey brains. This analysis was performed to examine whether the PCGS and the ILS were human-specific features. Towards that goal, each MRI scan was visually inspected to assess the presence or absence of these sulci in either hemisphere of each normalized brain for each species assessed.

(2) **A restricted qualitative/quantitative analysis** of all sulci of the MFC in both hemispheres of a subset of 40 brains of each species.

First, this analysis consisted in assessing the characteristics of the vertical sulci in the MFC emerging from the CGS or the PCGS in all brains across species. Towards that goal, the sulci were assigned in one of 3 categories based on our qualitative observations (Supplementary Fig. 1): sulcus (a real sulcus—long and deep—can be observed), spur (a precursor sulcus, i.e., not long enough to be considered a sulcus), and dimple (locations where a slight indentation indicates a dimple of the CGS or PCGS).

Second, we identified all sulci in the MFC and assessed any relationships that may exist between these sulci and certain fixed anatomical landmarks across the species examined, such as the rostral limit of the pons, anterior commissure, caudal and rostral limits of the genu of the corpus callosum. The relationships between the location of a given sulcus and a particular anatomical landmark across species were examined as follows:

(a) To assess whether PACS, PRPACS, VPCGS-P, and VPCGS-A were located at the level of, respectively, the most rostral limit of the pons (landmark 1), the anterior commissure (landmark 2), the caudal (landmark 3), and the rostral (landmark 4) limit of the genu of the corpus callosum, we calculated the difference between the $Y$ value of the intersection between the CGS or the PCGS (if present) and the PACS, PRPACS, VPCGS-P, and VPCGS-A and the $Y$ value of landmarks 1, 2, 3, and 4, respectively. This difference was calculated in all four species on the normalized T1 data of these species. This difference was then normalized to take into account the different antero-posterior extent of the brains of the four species, the antero-posterior extent of the human, chimpanzee, baboon, and macaque brains being respectively 175, 110, 85, and 60 mm. The normalization performed within species was obtained by dividing the $Y$ coordinate of the sulcus of interest, measured on brains registered linearly in the species-specific standard space, by the antero-posterior extent of the standard brain.

(b) To assess the relative location of the intersection of the fork formed by SU-ROS/CGS-VE and SOS/CGS-DE with the CGS or PCGS across primates, we calculated the % of displacement of this intersection from the rostral limit of the genu of the corpus callosum. Towards that goal, we measured the difference between the $Z$ value of this intersection and the $Z$ value of the rostral limit of the genu of the corpus callosum. To compare primate brains, this difference was then

normalized on the basis of the dorso-ventral extent of the brains normalized in their respective standard space (see "Methods") at the level of the rostral limit of the genu of the corpus callosum across primates (i.e., 100, 50, 30, and 24 mm, respectively, in human, chimpanzee, baboon, and macaque brains).

In both analyses, C.A. labeled the sulci. The labeling was performed on three separate occasions, and the few remaining inconsistencies in labeling were discussed with J.S. and M.P. Regarding the human data, validation of the nomenclature used was obtained in an independent experiment in which the naming of the sulci was performed by another expert[34].

**Statistical analysis**. We tested the influence of species status on the probability of a sulcus to be present with logistic regressions. In the statistical models, "species" (human, chimpanzee, baboon, macaque) was the independent variable and "Pre-sence" (0, 1) of a sulcus was the dependent variable. The lateralization of PCGS and ILS in human versus chimpanzee was assessed separately in each species using a binomial logistic regression GLMs with hemisphere (left versus right) as inde-pendent variable. To assess whether the probability of observing the four vertical sulci (i.e., PACS, PREPACS, VPCGS-P, and VPCGS-A) in the dorsal MFC was similar or different in each species, we performed binomial logistic regression GLMs in each species with these sulci (PACS, PREPACS, VPCGS-P, and VPCGS-A) as independent variable. ANOVA Chi-square tests and post-hoc Tukey tests were then applied. Note that, in the cases where only one value was observed in a specific variable (e.g., the PCGS is absent in baboon and macaque and therefore the "presence" value is 0 for all subjects), the binomial logistic regression GLM was fitted using an adjusted-score approach to bias reduction (using the brglm package; https://cran.r-project.org/web/packages/brglm/brglm.pdf).

In addition, we tested whether the relative distributions of $Z$ positions of the fork formed by SU-ROS/CGS-VE, SOS/CGS-DE, and the CGS or PCGS (when present), relative to the rostral limit of the genu of the corpus callosum were similar across species using standard generalized linear models in which the independent variable was "species" (human, chimpanzee, baboon, macaque) and the dependent variable was the normalized $Z$ relative position of this fork. This displacement was normalized across primates on the basis of the dorso-ventral extent of the brains normalized in their respective standard space at the level of the rostral limit of the genu of the corpus callosum (i.e., 100, 50, 30, and 24 mm, respectively, in human, chimpanzee, baboon, and macaque brains). The normalization performed within species was obtained by dividing the $Z$ coordinate of the intersection between the fork and the CGS or PCGS, measured on brains registered linearly in the species-specific standard space, by the dorso-ventral extent of the standard brain. All data are presented in the Source Data file.

All statistics were performed with R software, R Development Core Team[69] under R-Studio[70].

**Reporting summary**. Further information on research design is available in the Nature Research Reporting Summary linked to this article.

## Data availability

The source data underlying Figs. 1e, 2a, b, 3b–e, 4a–d, 5a–c, 6a–c, 7a, b, and Supplementary Fig. 2a, b are provided as a Source Data file. Human, macaque, chimpanzee, and baboon anatomical scans are available, respectively, from the Human Connectome Project database [http://www.humanconnectome.org/], the PRIMatE Data Exchange (PRIME-DE) database [https://fcon_1000.projects.nitrc.org/indi/indiPRIME.html], from Dr. W. Hopkins [http://www.chimpanzeebrain.org/], and from Dr. A. Meguerditchian [http://www.nitrc.org/projects/haiko89/]. A reporting summary for this Article is available as a Supplementary Information file.

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

## Acknowledgements

This work was supported by the Human Frontier Science Program (RGP0044/2014). C. A. received funding from the French National Research Agency (ANR-18-CE32-0012-01). J.S. was supported by a Sir Henry Dale Wellcome Trust Fellowship (105651/Z/14/Z). The Wellcome Centre for Integrative Neuroimaging is supported by core funding from the Wellcome Trust (203139/Z/16/Z). W.D.H. is supported by NIH grants NS-042867, NS-073134, and NS-092988. A.M. has received funding from the European Research Council (ERC) under the European Union's Horizon 2020 research and innovation program Grant Agreement No. 716931 (716931 – GESTIMAGE – ERC-2016-STG), from the French "Agence Nationale de la Recherche" (ANR-12-PDOC-0014-01, LangPrimate Project), as well as from grants ANR-16-CONV-0002 (ILCB), ANR-11-LABX-0036 (BLRI), and ANR-11-IDEX-0001-02 (A*MIDEX). F.H.-B. received funding from the French National Research Agency (ANR-15-CE37-0003). E.P. was supported by the Medical Research Foundation (FRM), Neurodis Foundation, and the Labex CORTEX ANR-11-LABX-0042 of Université de Lyon. M.P. was supported by the Canadian Institutes of Health Research (CIHR) Foundation grant FDN-143212. E.P., C.A., F.H.-B., and S.B.H. are employed by the Centre National de la Recherche Scientifique. The authors thank K. Knoblauch for helpful comments on statistical analysis. The authors thank Delphine Autran-Clavagnier for data acquisition in macaques.

## Author contributions

C.A. organized the project, coordinated the consortium to gather the dataset, and analyzed the data. C.A., J.S. and M.P. interpreted data and wrote the article. W.D.H. provided chimpanzee and macaque T1 data; A.M. provided baboon T1 data; F.H.-B., S.B.H., J.S., W.D.H., C.R.E.W., C.A. and E.P. provided macaque T1 data.

## Additional information

**Competing interests:** The authors declare no competing interests.

