## [Peer Review File · Nature Communications]

Reviewers' Comments:

Reviewer #1:

Remarks to the Author:

Comparative anatomy is a method to infer about evolutionary processes.

When applied to the brain, traditionally anatomists compare the relative size of any structures across species and seek for unique features existing solely in humans.

In this context, Amiez et al. through a multinational effort, assessed the sulcal organisation of the medial frontal cortex in 4 closely related primates species. This is, to my knowledge, the largest comparative anatomy study using neuroimaging. The results are original and consist in the discovery that (1) paracingulate sulcus does not only exist in humans but also in chimpanzees (2) homologies exist between the 4 species within the medial frontal cortex (3) the rostral part of the frontal lobe expanded along the phylogeny scale.

This is an excellent anatomical work and a solid contribution to the field of neuroanatomy, and brain evolution models. As correctly stated by the authors, this study provides a solid framework for future investigations to come.

A few points caught my attention.

The rationale of the introduction includes the statement that main sulci are boundaries to cytoarchitectonic areas and functional territories.

"For instance, the central sulcus is the limiting sulcus between the primary motor cortex (area 4) and the primary somatosensory cortex (area 3)" is it really? looking at the work derived from per-operative electrical stimulation the motor cortex appears to extend beyond the central sulcus (Penfield and Boldrey, 1937). Similarly, recent cytoarchitectonic maps (area 4) produced by the Jülich Group extend beyond the central sulcus as well. Such a strong statement needs to be clarified as the rest of the argument is focused on functional territories localised using sulci.

I noticed a contradiction in the result section. This needs to be edited.

'presence of the PCGS in at least one hemisphere within human and chimpanzee brains' suggest that PCGS is always at least in one hemisphere, however, the authors further wrote 'PCGS in least in one hemisphere in 70.1% of human versus 33.8% of chimpanzee brains'

This seems to be contradictory and needs editing.

The remaining of the paper is excellent. The method is spotless. The discussion was solid and quite enjoyable, however a bit heavy for the lay reader.

Reviewer #2:

Remarks to the Author:

Amiez, Petrides and colleagues revisit the sulcal organization to provide a new understanding into evolution of primate frontal cortex. They discovered that both Hominoidea do possess a paracingulate sulcus, which was previously thought to be uniquely human and linked to higher cognitive functions like mentalizing; using neuroimaging T1 anatomical data of 197 humans, 225 chimpanzees, 88 baboons and 80 macaque brains were analyzed. The topic of the specially human features of the frontal lobe is of continued interest and thus relevant to the wide readership of Nature Communications. I highly recommend this fine piece of empirical science for publication, after incorporating changes to the following suggestions:

Major

a) Introduction: ...could be slightly streamlined a little more. In particular, the 2nd paragraph contains a series of examples and may be shorted by a few lines.

b) Methods: Although openly available datasets are used for the human and monkey data, it will be useful for the manuscript to stand for itself and be reproducible to provide more details on the preprocessing procedures for the T1 data. Especially with regards to those preprocessing steps that may have influenced the commonality vs. divergence between the species in sulcus architecture. This information would be necessary to judge the adequacy of the findings and conclusions.

c) Methods: The main analyses are described by "we built binomial logistic regression GLMs." From this very brief statement it is hard to know or reproduce what was actually done in the work. For instance, which input variables and what dependent variable were fed into the model. Were any potential confounds or interest removed, and were the measures z-scored besides other relevant details.

d) Results: As a general suggestion, in the results section it may be useful after sentence like "We also tested for asymmetries in the presence of ILS within each species." to unpack who the substantive question of asymmetries was exactly quantitatively addressed based on the data. This would also help for readers who decide go right right to the results after reading the introduction, skipping the methods section.

e) Some statements be left loose, such as "These abilities reach their epitope in humans, strongly suggesting that these cortical areas have evolved. The present analysis reveals how these regions have evolved, providing a structural substrate for improved abilities in the primate order." It may be helpful to be more explicit what order of evolutionary appearance of the anatomical sulci in question may be according to these authors.

f) Limitations: ...are perhaps not fully acknowledged yet in the discussion of the manuscript. This reviewer recommends adding a limitations paragraph; especially to remind and make explicit how the sulci were annotated in the brain scans and how these annotations were then fed into statistical analyses.

Minor

a) Figures: the abbreviations, such as DMPS or VMPS, are sometimes not explained in the figure caption. This addition could however be useful to make each figure more autonomous.

b) Considering rephrasing/simplifying: "The present analysis adds up on these results revealing that"

c) Style: the paragraph starting out by "The most ventral sulcus within the vmPFC " appears unaesthetically short. Perhaps integrate with the other paragraphs for coherence.

Reviewer #3:

Remarks to the Author:

The study by Amiez and colleagues is extremely impressive in the care with which a substantial amount of MRI structural data in 4 species of primates (macaque, baboon, human and chimpanzee) is

expertly neuroanatomically compared with regards to medial frontal cortex sulcal organization.

Overall I find the work to be expertly conducted and although I initially had a concern that some of the results are correlated with brain size, the MS has important internal controls, meaning that there is evidence for selective higher order gyrification in humans and some other species for some structures but not others which cannot be easily explained simply by non-selective higher order gyrification for bigger brains. But the discussion could be stronger in this regard. I also wonder if there is a way to add evolutionary time separating the species to some of the key analyses which again will help to weigh results that cannot be primarily by brain size differences. I'm not sure in that regard because it is just 4 evolutionary distance timepoints, so will leave it with the authors to decide. However, it occurred to me that the macaque/baboon distinction is important because they have different brain sizes but presumably not vastly different common ancestor split dates from the other species. So there is interesting evolutionary information to better guide discussion of the summary figure (Fig. 8).

Here are my comments in more detail:

1. The extent to which the results relate to non-selective higher order gyrification in bigger brains needs to be more convincingly ruled out in the discussion. You start off the discussion with noting that there is typically a correlation in such cross-species comparisons but then make no further comments. I would make sure that you are able to conceptually weigh the results more when there is evidence for selective higher order gyrification that cannot be explained by larger brains from macaques, to baboons, to chimps to humans. So a better discussion in reference to your internal controls (MCC results or the like that are very similar across all species) could help to strengthen the manuscript.
2. Fig. 8. I am unclear with the summary figure what the suggested evolutionary picture between macaques and baboons is given that they are both Old World monkeys, yet have very different brain sizes. Are you suggesting they faced different evolutionary pressures, if so please comment further. More care in discussing the results that cannot be attributed to brain size differences would be good. Even better would be if at least some of your key analyses can consider the evolutionary time separating the species, as far as it is known. But I understand that might be challenging to do and so I'll leave it up to you whether that would be a useful and informative additional analysis component, again only for some of the key results. I can understand that a co-variable with just four evolutionary dates won't be particularly powerful, but at least conceptually for the discussion it could be.
3. Methods: there is little information on who did the delineation of the brain for this study. I can appreciate that you need an anatomical expert, but at the same time to reduce potential subconscious bias it would have been good if this was done by someone who was blind to the hypotheses. Can you say more about the scorers and how replicable the results are if multiple scorers. The brains across the species are vastly different so one cannot be blind to the species.
4. Discussion: Is there any clear functional basis that can be pointed to for the human left hemisphere asymmetry in the PCGS? I presume this is not known but if it is could benefit from a bit more information in the discussion.
5. First sentence in introduction: "Is the human brain unique?" should be "How is the human brain unique?". All brains are unique in their own ways.
6. Last paragraph on p. 5 is non sequitur: "Opposing views suggest that the MFC is functionally comparable in human and macaque". They can't be opposing if they suggest the same thing.

Responses to reviewers

Reviewer #1:

Comparative anatomy is a method to infer about evolutionary processes. When applied to the brain, traditionally anatomists compare the relative size of any structures across species and seek for unique features existing solely in humans. In this context, Amiez et al. through a multinational effort, assessed the sulcal organisation of the medial frontal cortex in 4 closely related primates species. This is, to my knowledge, the largest comparative anatomy study using neuroimaging. The results are original and consist in the discovery that (1) paracingulate sulcus does not only exist in humans but also in chimpanzees (2) homologies exist between the 4 species within the medial frontal cortex (3) the rostral part of the frontal lobe expanded along the phylogeny scale. This is an excellent anatomical work and a solid contribution to the field of neuroanatomy, and brain evolution models. As correctly stated by the authors, this study provides a solid framework for future investigations to come.

We thank the reviewer for these very positive comments.

A few points caught my attention.

The rationale of the introduction includes the statement that main sulci are boundaries to cytoarchitectonic areas and functional territories. "For instance, the central sulcus is the limiting sulcus between the primary motor cortex (area 4) and the primary somatosensory cortex (area 3)" is it really? looking at the work derived from per-operative electrical stimulation the motor cortex appears to extend beyond the central sulcus (Penfield and Boldrey, 1937). Similarly, recent cytoarchitectonic maps (area 4) produced by the Jülich Group extend beyond the central sulcus as well. Such a strong statement needs to be clarified as the rest of the argument is focused on functional territories localised using sulci.

We thank the reviewer for pointing out that our statement requires further clarification because it could potentially be misinterpreted. We have re-written the relevant paragraph of the manuscript to make clear what we mean by 'limiting sulcus'. Clearly, as the reviewer points out, the motor cortex (area 4) which lies on the anterior bank of the central sulcus extends to a variable degree onto the surface of the precentral gyrus. Similarly, the primary somatosensory cortex area 3 which lies on the posterior bank of the central sulcus may extend to a limited extent onto the crown of the postcentral gyrus. By using the word "limiting" we meant to state that the anterior bank of the central sulcus is occupied by primary motor cortex and the posterior bank by primary somatosensory cortex and thus the central sulcus separates the motor from the somatosensory cortical regions. This observation was made by Brodmann (1909) and was more recently confirmed by White and colleagues (White et al. 1997). These investigators examined the morphology and cytoarchitecture of the central sulcus in 20 human brains. They observed that, although the surface appearance of the central sulcus varies greatly from brain to brain (and between hemispheres in individual brains), in the Nissl stained sections of the central sulcus that were cut orthogonal to the direction of the sulcus (i.e. sections optimal for cytoarchitectonic analysis), the locations of the borders of area 4 (primary motor cortex) and area 3

(somatosensory) along the course of the sulcus were similar among the 40 hemispheres examined. Thus, the central sulcus divided the motor cortex from the somatosensory region. Also, Geyer et al (1999) from the Juelich group who examined cytoarchitectonically areas 3a, 3b and 1 in 10 cadaver brains noted that, although there was a certain degree of variability in the relation of the border of these areas and macrostructural features across individual brains, area 3a was lying in the fundus of the central sulcus, area 3b in the rostral bank of the postcentral gyrus (i.e. the posterior bank of the central sulcus), and area 1 was observed on the crown of the postcentral gyrus.

Based on the reviewer's comment, we have re-written in the revised manuscript the relevant part of the paragraph to clarify this point. By "limiting" we mean that the central sulcus separates the motor from the somatosensory region, we do not mean that the functional and cytoarchitectonic area 4 does not extend beyond the anterior bank of the central sulcus onto the crown of the precentral gyrus to a variable extent. It does extend onto the precentral gyrus both functionally (Penfield's work) and cytoarchitectonically. By limiting we meant to say that the anterior bank of the central sulcus is occupied by motor cortical area 4 and the posterior bank by somatosensory area 3 and thus the central sulcus is a dividing line between the motor cortical region and the somatosensory cortical region and such organization does not preclude some inter-individual variability in the relation of the precise border of these areas across individual brains. We expect that in the revised relevant paragraph this point is made clear and we also emphasize that we do not think that ALL of area 4 is in the anterior bank of the central sulcus. We thank the reviewer for pointing out this potential mis-understanding and asking us to clarify what we mean by 'limiting' sulcus.

To also comply with reviewer 2's request regarding the length of our introduction, we only provided the reader with a summarized version of the previous paragraph. Here is the revised text in the relevant paragraph:

"Importantly, several lines of evidence indicate that primary sulci are limiting sulci between cytoarchitectonic areas. For instance, the central sulcus is the limiting sulcus between the primary motor cortex (area 4) which occupies the anterior bank of the sulcus and extends for a variable distance on the precentral gyrus and the primary somatosensory cortex (area 3) that lies on its posterior bank²²⁻²⁶. Although there is some inter-individual variability in the relation of the precise border of these areas and macrostructural features across individual brains²², the central sulcus remains a dividing line between the motor cortical region, anteriorly, and the somatosensory cortical region, posteriorly^{e.g. 22,25,27,28}."

I noticed a contradiction in the result section. This needs to be edited. 'presence of the PCGS in at least one hemisphere within human and chimpanzee brains' suggest that PCGS is always at least in one hemisphere, however, the authors further wrote 'PCGS in least in one hemisphere in 70.1% of human versus 33.8% of chimpanzee brains'

This seems to be contradictory and needs editing.

We meant that a PCGS can be observed at least in one hemisphere in 70.1% of human and 33.8% of chimpanzee. To clarify this important point, we rephrased the first paragraph of the result section. It now reads:

“A major finding of the present analysis was the demonstration that a PCGS can be observed, at least in one hemisphere, in 70.1% of human brains and 33.8% of chimpanzee brains, but not in baboon and macaque brains (dependent variable: PCGS present (0/1), main effect species: $\chi^2 = 242.18$, $df = 3$, $p\text{-value} = 2.2e\text{-}16$, logistic regression, GLM fitted using an adjusted-score approach to bias reduction) (Fig 1A, B, C, D, E). A post-hoc Tukey test demonstrated that the probability of occurrence of a PCGS is significantly decreased from human to chimpanzee brains (estimate=1.47562, std. error=0.20917, $z\text{-value}=7.054$, $p\text{-value}<0.001$) (Fig 1E).”

The remaining of the paper is excellent. The method is spotless. The discussion was solid and quite enjoyable, however a bit heavy for the lay reader.

We thank the reviewer for these very positive comments. We hope that the discussion is now improved thanks to all reviewers' comments.

Reviewer #2

Amiez, Petrides and colleagues revisit the sulcal organization to provide a new understanding into evolution of primate frontal cortex. They discovered that both Hominoidea do possess a paracingulate sulcus, which was previously thought to be uniquely human and linked to higher cognitive functions like mentalizing; using neuroimaging T1 anatomical data of 197 humans, 225 chimpanzees, 88 baboons and 80 macaque brains were analyzed. The topic of the specially human features of the frontal lobe is of continued interest and thus relevant to the wide readership of Nature Communications. I highly recommend this fine piece of empirical science for publication, after incorporating changes to the following suggestions:

We thank the reviewer for these positive comments.

Major

a) Introduction: ...could be slightly streamlined a little more. In particular, the 2nd paragraph contains a series of examples and may be shorted by a few lines.

We shortened the 2d paragraph. It now reads:

“Sulcal organization is not random despite the presence of strong inter-hemispheric and inter-subject variability ¹⁴. Rather, it follows a precise topographical organization. A straightforward example is the location of the central sulcus. Although its shape, length, and depth may vary across hemispheres and individuals, it is always present and systematically located at the same strategic antero-posterior location. Importantly, this is not a human specific feature as it is observed in all primates ^{15,16}. Although the origin of gyrification is not well understood, three types of sulci can be identified in primates, based on their appearance during gestation. Whereas primary sulci, which appear first during gestation (e.g. central sulcus, cingulate sulcus –CGS–) ^{17,18}, are present in all hemispheres in all individuals, the probability of observing secondary/tertiary sulci is variable. For example, the paracingulate sulcus (PCGS) is present only in about 70% of subjects at least in one hemisphere ^{9,19–21}.”

b) Methods: Although openly available datasets are used for the human and monkey data, it will be useful for the manuscript to stand for itself and be reproducible to provide more details on the preprocessing procedures for the T1 data. Especially with regards to those preprocessing steps that may have influenced the commonality vs. divergence between the species in sulcus architecture. These information would be necessary to judge the adequacy of the findings and conclusions.

First, we would like to emphasize that the preprocessing of T1 data included only the linear registration of primate brains to their respective standard brain (e.g. each chimpanzee T1 data are registered in the chimpanzee standard brain). Importantly, such normalization procedure (using species specific standard spaces) has the great advantage of allowing within-species comparisons between brains without changing relationships between sulci and gyri. It is therefore very unlikely that such processing influenced commonality and divergence of sulcal organization observed between species. We now clarified this point in the Method section and thank the reviewer for raising this issue.

The relevant paragraph in the Method section now reads:

*“**Neuroimaging data analysis.** Human and macaque brains were normalized in the human (<http://www.bic.mni.mcgill.ca/ServicesAtlases/HomePage>) and macaque⁹⁴ MNI stereotaxic coordinate systems, respectively. Chimpanzee brains were normalized in the chimpanzee standard brain developed by Dr. W. Hopkins⁹⁵ (www.chimpanzeebrain.org). Baboon brains were normalized in the baboon standard brain developed by Dr. A. Meguerditchian⁹⁶ (<http://www.nitrc.org/projects/haiko89/>). Note that normalization of all primate brains consisted in linear registrations, which has the great advantage of allowing within-species comparisons between brains without altering relationships between sulci and gyri. It is therefore unlikely that such processing influences commonality and divergence of sulcal organization observed between species. Normalization of primate brains was performed with SPM12 (<https://www.fil.ion.ucl.ac.uk/spm/software/spm12/>).”*

Second, the reproducibility of data will be ensured by the fact that all T1 data are open source and that files containing all the data are now provided.

c) Methods: The main analyses are described by "we built binomial logistic regression GLMs." From this very brief statement it is hard to know or reproduce what was actually done in the work. For instance, which input variables and what dependent variable were fed into the model. Were any potential confounds or interest removed, and were the measures z-scored besides other relevant details.

We re-phrased this part of the text. We “performed” binomial logistic regressions to test whether the probability of occurrence of a sulcus varied between species. We now indicate in each statistical test presented in the result section the independent and dependent variables used in these models.

We also rephrased the Method section. It now reads:

“We tested the influence of species status on the probability of a sulcus to be present with logistic regressions. In the statistical models, ‘species’ (Human, Chimpanzee, Baboon, Macaque) was the independent variable and ‘Presence’ (0, 1) of a sulcus was the dependent variable. The lateralization of PCGS and ILS in human versus chimpanzee was assessed

separately in each species using a binomial logistic regression GLMs with hemisphere (left versus right) as independent variable. To assess whether the probability of observing the four vertical sulci (i.e. PACS, PREPACS, VPCGS-P, and VPCGS-A) in the dorsal medial frontal cortex was similar or different in each species, we performed binomial logistic regression GLMs in each species with these sulci (PACS, PREPACS, VPCGS-P, and VPCGS-A) as independent variable. ANOVA Chi-square tests and post-hoc Tukey tests were then applied. Note that, in the cases where only one value was observed in a specific variable (e.g. the PCGS is absent in baboon and macaque and therefore the “presence” value is 0 for all subjects), the binomial logistic regression GLM was fitted using an adjusted-score approach to bias reduction (using the *brglm* package (<https://cran.r-project.org/web/packages/brglm/brglm.pdf>)).

In addition, we tested whether the relative distributions of Z positions of the fork formed by SU-ROS/CGS-VE, SOS/CGS-DE, and the CGS or PCGS (when present), relative to the rostral limit of the genu of the corpus callosum were similar across species using standard generalized linear models in which the independent variable was ‘species’ (Human, Chimpanzee, Baboon, Macaque) and the dependent variable was the normalized Z relative position of this fork. This displacement was normalized across primates on the basis of the dorso-ventral extent of the brains normalized in their respective standard space at the level of the rostral limit of the genu of the corpus callosum (i.e. 100mm, 50mm, 30mm, and 24mm, respectively, in human, chimpanzee, baboon, and macaque brains). The normalization performed within species was obtained by dividing the Z coordinate of the intersection between the fork and the CGS or PCGS, measured on brains registered linearly in the species-specific standard space, by the dorso-ventral extent of the standard brain. All data are presented in Dataset 1. All statistics were performed with R software, R Development Core Team⁹⁷ under R-Studio⁹⁸.

d) Results: As a general suggestion, in the results section it may be useful after sentence like "We also tested for asymmetries in the presence of ILS within each species." to unpack who the substantive question of asymmetries was exactly quantitatively addressed based on the data. This would also help for readers who decide go right to the results after reading the introduction, skipping the methods section.

To clarify this point, we rephrased the sentence (Result section p. 8). It now reads:

“We also tested for asymmetries in the presence of ILS within each species by assessing whether the ILS was more frequent in one hemisphere compared to the other. No interhemispheric differences were found in humans (dependent variable: ILS present (0/1), main effect hemispheres (Left/Right): $\chi^2 = 0.68999$, $df = 1$, $p\text{-value} = 0.4062$ (ns), baboons ($\chi^2 = 3.3591$, $df = 1$, $p\text{-value} = 0.06684$ (ns), or rhesus monkeys ($\chi^2 = 0.42938$, $df = 1$, $p\text{-value} = 0.5123$ (ns), logistic regression). However, the ILS was found to be present significantly more often in the left than the right hemisphere in chimpanzees ($\chi^2 = 10.419$, $df = 1$, $p\text{-value} = 0.001247$, logistic regression). Finally, in Hominoidea, when a PCGS was present, the ILS was almost always absent (Fig 2B) (Human: presence of ILS in hemispheres with a PCGS versus hemispheres without a PCGS: $\chi^2 = 30.221$, $df = 1$, $p\text{-value} = 3.855e-08$; Chimpanzee: $\chi^2 = 14.011$, $df = 1$, $p\text{-value} = 0.0001817$, logistic regression).”

e) Some statements be left loose, such as "These abilities reach their epitope in humans, strongly suggesting that these cortical areas have evolved. The present analysis reveals how these regions have evolved, providing a structural substrate for improved abilities in

the primate order." It may be helpful to be more explicit what order of evolutionary appearance of the anatomical sulci in question may be according to these authors.

We understand the reviewer is asking about the causal link between brain structure and function from an evolutionary perspective. Although it is difficult to make such claims notably because of the limited number of species studied (n=4) we modified the last paragraphs of our discussion to propose our interpretation of the issue raised by the reviewer:

"Altogether comparative neuroanatomy studies show a relatively well conserved anatomo-functional organization of the medial prefrontal cortex in primates^{46,60,62}, which suggests that the precursor of the human brain was already present in the last ancestor to hominids, apes and old-world monkeys. Differences in sulcal pattern morphology as observed in the present study could be related to the expansion of cortical areas, which in turn could be related to expansion in motor and cognitive processes. Importantly, the brain size difference across primate species does not explain by itself the pattern of sulcal organization observed in the present study as we observed 1) different patterns of sulcal morphology within each species (controlling therefore for brain size), and 2) a consistent antero-posterior organization of the vertical sulci and their precursors in the dorsal medial frontal cortex in the four species (i.e. in different brains of different sizes). From studies assessing within species inter-individual variability in sulcal morphology and behavioral performance, we know of the existence of relationships between structure and function. Such evidence exists in human^{e.g. 81-84}, but also in chimpanzee brains^{e.g. 85}. Indeed, Hopkins et al. (2017)⁸⁵ found that males producing attention getting sounds showed a larger leftward asymmetry in the depth of central and ventral portion of the central sulcus compared to males that produce less attention getting sounds whereas females show the opposite pattern. We also recently showed that the presence of a PCGS and an intralimbic sulcus, particularly in the left hemisphere, was associated respectively with the production and use of attention-getting sounds by chimpanzees and rightward handedness (Hopkins et al. in preparation). It is therefore reasonable to think that interindividual variability in sulcal morphology in macaque and baboon will be associated with variability in behavioural performance.

The present study provides critical new evidence in the context of a new comprehensive framework regarding prefrontal cortical evolution. On one hand, these results are in agreement with previous comparative connectomic studies^{46,62} and show a conserved organization of the medial prefrontal cortex, suggesting the existence of a precursor architecture of the human brain in the last common ancestor to humans, apes, and old-world monkeys. On the other hand, the appearance in human and ape brains of the PCGS, and the relative expansion of the rostral and dorsal part of the medial prefrontal cortex in humans (emergence of the RPPCGS and the more ventral position of the intersection between SUROS/CGS-VE, SOS/CGS-DE and CGS) (summarized in Fig 8) might reflect the evolution of mentalizing processes from a rudimentary form in Old-World monkeys^{86,87}, to a more complex form in apes⁸⁸, and in humans⁸⁹."

f) Limitations: ...are perhaps not fully acknowledged yet in the discussion of the manuscript. This reviewer recommends adding a limitations paragraph; especially to remind and make explicit how the sulci were annotated in the brain scans and how these annotations were then fed into statistical analyses.

We now added a paragraph in the Discussion about limitations. This paragraph reads:

“Manual labelling of the sulci following visual inspection in the various species in the present study may appear as a limitation. Yet, a similar strategy has been recently employed and confirmed the organization of these sulci in humans⁵⁶. Also, we are convinced that such precise analysis of sulci in multiple species represents an important step toward the development of efficient algorithms for automatic labelling, which are currently not available. To characterize further the evolution of sulcal patterns in primates, future studies will need to consider the assessment of more species. But, as no endocast exists for the medial frontal cortex, it is important to note that only analysis on living species could be conducted to understand how structures on the medial surface of the brain have evolved.”

Minor

a) Figures: the abbreviations, such as DMPS or VMPS, are sometimes not explained in the figure caption. This addition could however be useful to make each figure more autonomous.

We modified the legends accordingly.

b) Considering rephrasing/simplifying: "The present analysis adds up on these results revealing that"

We rephrased this sentence. It now reads: *“The present analysis shows that the sulcal organization of this region is also well-preserved.”*

c) Style: the paragraph starting out by "The most ventral sulcus within the vmPFC " appears unaesthetically short. Perhaps integrate with the other paragraphs for coherence.

We rephrased this sentence and integrated it to the preceding paragraph.

Reviewer #3

The study by Amiez and colleagues is extremely impressive in the care with which a substantial amount of MRI structural data in 4 species of primates (macaque, baboon, human and chimpanzee) is expertly neuroanatomically compared with regards to medial frontal cortex sulcal organization. Overall I find the work to be expertly conducted and although I initially had a concern that some of the results are correlated with brain size, the MS has important internal controls, meaning that there is evidence for selective higher order gyrification in humans and some other species for some structures but not others which cannot be easily explained simply by non-selective higher order gyrification for bigger brains. But the discussion could be stronger in this regard. I also wonder if there is a way to add evolutionary time separating the species to some of the key analyses which again will help to weigh results that cannot be primarily by brain size differences. I'm not sure in that regard because it is just 4 evolutionary distance timepoints, so will leave it with the authors to decide. However, it occurred to me that the macaque/baboon distinction is important because they have different brain sizes but presumably not vastly

different common ancestor split dates from the other species. So there is interesting evolutionary information to better guide discussion of the summary figure (Fig. 8).

We thank the reviewer for these very positive comments. We respond below to all specific comments.

Here are my comments in more detail:

1. The extent to which the results relate to non-selective higher order gyrification in bigger brains needs to be more convincingly ruled out in the discussion. You start off the discussion with noting that there is typically a correlation in such cross-species comparisons but then make no further comments. I would make sure that you are able to conceptually weigh the results more when there is evidence for selective higher order gyrification that cannot be explained by larger brains from macaques, to baboons, to chimps to humans. So a better discussion in reference to your internal controls (MCC results or the like that are very similar across all species) could help to strengthen the manuscript.

We fully agree with the reviewer that brain size is an important factor when considering gyrification issues. However new sulci do not appear uniformly in the MFC; those sulci are in regions that have expanded more in humans than in monkeys.

The more straightforward demonstration that brain size difference does not explain by itself the pattern of sulcal organization observed in the present study is that, within each species (controlling therefore for brain size), we observed different patterns of sulcal morphology. One emerging question is whether non-human primates displaying different sulcal patterns behave differently, as observed in human (e.g. Buda et al. 2011, Mériaux et al. 2006, Cachia et al. 2016, Borst et al. 2014). Recent evidence suggests that it is the case also in chimpanzee (e.g. Hopkins et al. 2017). Indeed, Hopkins et al. (2017) found that males producing attention getting sounds showed a larger leftward asymmetry in the depth of central and ventral portion of the central sulcus compared to males that produced less attention getting sounds whereas females show the opposite pattern. We also recently showed that the presence of a PCGS and an intralimbic sulcus, particularly in the left hemisphere, was associated respectively with the production and use of attention-getting sounds by chimpanzees and rightward handedness (Hopkins et al. in preparation).

We added the following paragraphs in the Discussion section to clarify this point:

“Although gyrification correlates with brain volume ^{2,57}, and therefore with the expansion of primate neocortex, the present results did not reveal random or homogeneous changes in sulcal morphology in the medial prefrontal cortex across the four primate species examined (e.g. differences in the ACC/vmPFC but no differences in the MCC/medial premotor cortex). The observed differences were in specific regions that have expanded more during evolution ⁵⁸.”

“As we observed reduced variability in the ACC/vmPFC of humans and chimpanzees compared to Old World monkeys, we suggest that a facing downward SU-ROS/CGS-VE morphology might be associated with a trait that is conferring an evolutionary advantage, and was therefore selected.

“Manual labelling of the sulci following visual inspection in the various species in the present study may appear as a limitation. Yet, a similar strategy has been recently employed and confirmed the organization of these sulci in humans⁵⁶. Also, we are convinced that such precise analysis of sulci in multiple species represents an important step toward the development of efficient algorithms for automatic labelling, which are currently not available. To characterize further the evolution of sulcal patterns in primates, future studies will need to consider the assessment of more species. But, as no endocast exists for the medial frontal cortex, it is important to note that only analysis on living species could be conducted to understand how structures on the medial surface of the brain have evolved.

Altogether comparative neuroanatomy studies show a relatively well conserved anatomo-functional organization of the medial prefrontal cortex in primates^{46,60,62}, which suggests that the precursor of the human brain was already present in the last ancestor to hominids, apes and old-world monkeys. Differences in sulcal pattern morphology as observed in the present study could be related to the expansion of cortical areas, which in turn could be related to expansion in motor and cognitive processes. Importantly, the brain size difference across primate species does not explain by itself the pattern of sulcal organization observed in the present study as we observed 1) different patterns of sulcal morphology within each species (controlling therefore for brain size), and 2) a consistent antero-posterior organization of the vertical sulci and their precursors in the dorsal medial frontal cortex in the four species (i.e. in different brains of different sizes). From studies assessing within species inter-individual variability in sulcal morphology and behavioral performance, we know of the existence of relationships between structure and function. Such evidence exists in human^{e.g. 81–84}, but also in chimpanzee brains^{e.g. 85}. Indeed, Hopkins et al. (2017)⁸⁵ found that males producing attention getting sounds showed a larger leftward asymmetry in the depth of central and ventral portion of the central sulcus compared to males that produce less attention getting sounds whereas females show the opposite pattern. We also recently showed that the presence of a PCGS and an intralimbic sulcus, particularly in the left hemisphere, was associated respectively with the production and use of attention-getting sounds by chimpanzees and rightward handedness (Hopkins et al. in preparation). It is therefore reasonable to think that interindividual variability in sulcal morphology in macaque and baboon will be associated with variability in behavioural performance.

The present study provides critical new evidence in the context of a new comprehensive framework regarding prefrontal cortical evolution. On one hand, these results are in agreement with previous comparative connectomic studies^{46,62} and show a conserved organization of the medial prefrontal cortex, suggesting the existence of a precursor architecture of the human brain in the last common ancestor to humans, apes, and old-world monkeys. On the other hand, the appearance in human and ape brains of the PCGS, and the relative expansion of the rostral and dorsal part of the medial prefrontal cortex in humans (emergence of the RPPCGS and the more ventral position of the intersection between SUROS/CGS-VE, SOS/CGS-DE and CGS) (summarized in Fig 8) might reflect the evolution of mentalizing processes from a rudimentary form in Old-World monkeys^{86,87}, to a more complex form in apes⁸⁸, and in humans⁸⁹.”

2. Fig. 8. I am unclear with the summary figure what the suggested evolutionary picture between macaques and baboons is, given that they are both Old World monkeys, yet have very different brain sizes. Are you suggesting they faced different evolutionary pressures, if so please comment further. More care in discussing the results that cannot be attributed

to brain size differences would be good. Even better would be if at least some of your key analyses can consider the evolutionary time separating the species, as far as it is known. But I understand that might be challenging to do and so I'll leave it up to you whether that would be a useful and informative additional analysis component, again only for some of the key results. I can understand that a co-variable with just four evolutionary dates won't be particularly powerful, but at least conceptually for the discussion it could be.

First of all macaques and baboons faced different evolutionary pressures as they adapted to different ecological niches. However, how these forces impacted differently on these two species is yet unknown. This is a very interesting point that future studies will need to address. It is nonetheless clear from our results that the differences between baboon and macaque cannot be attributed solely to brain size difference as we observed 1) different patterns of sulcal morphology within each specie (controlling therefore for brain size), and 2) a consistent antero-posterior organization of the vertical sulci and their precursors in the dorsal medial frontal cortex in the four species (i.e. in different brains of different sizes). As described in our reply to the previous points of the reviewer, we have addressed the question of differences associated with brain size in our discussion.

We also agree that assessing the impact of evolutionary time separating the species is of great interest. As said by the reviewer, the fact that only 4 species are used prevent us to assess efficiently this point. We now added this point in our limitation section in the Discussion. The limitation section reads:

“Manual labelling of the sulci following visual inspection in the various species in the present study may appear as a limitation. Yet, a similar strategy has been recently employed and confirmed the organization of these sulci in humans⁵⁶. Also, we are convinced that such precise analysis of sulci in multiple species represents an important step toward the development of efficient algorithms for automatic labelling, which are currently not available. To characterize further the evolution of sulcal patterns in primates, future studies will need to consider the assessment of more species. But, as no endocast exists for the medial frontal cortex, it is important to note that only analysis on living species could be conducted to understand how structures on the medial surface of the brain have evolved.”

3. Methods: there is little information on who did the delineation of the brain for this study. I can appreciate that you need an anatomical expert, but at the same time to reduce potential subconscious bias it would have been good if this was done by someone who was blind to the hypotheses. Can you say more about the scorers and how replicable the results are if multiple scorers. The brains across the species are vastly different so one cannot be blind to the species.

We now added information in the Method section. CA labelled the sulci in human. The labelling was performed at 3 separate occasions inter-spaced by 1 to 3 weeks, and the few left inconsistencies in labelling was discussed with JS and MP. Concerning human data, we validated our sulci nomenclature in an independent experiment, where the sulci naming was performed by another researcher (Lopez Persem et al. 2019). Concerning non-human primates, the labelling was performed by CA and JS. The labelling of the few left problematic brains was then discussed with MP.

We added details about the labelling process in the Method section:

“In both analyses, C.A. labelled the sulci in the human brains. The labelling was performed on three separate occasions, and the few remaining inconsistencies in labelling were discussed with J.S. and M.P. Regarding the human data, validation of the nomenclature used was obtained in an independent experiment in which the naming of the sulci was performed by another expert⁵⁶.”

We also agree that manual labelling of the sulci is a limitation of the current study in the sense that no automatic accurate labelling algorithm is currently available. In fact, efficient algorithm for automatic labelling require first such precise analysis in multiple species as in the present paper. Our study may therefore help such development. We now mentioned this issue in the limitation paragraph in the Discussion section.

4. Discussion: Is there any clear functional basis that can be pointed to for the human left hemisphere asymmetry in the PCGS? I presume this is not known but if it is could benefit from a bit more information in the discussion.

The leftward asymmetry observed in the human brain seems to be a human trait under the influence of environmental factors such as in-womb environment and under the weak influence of genetic factors (Amiez et al. 2018). Although its functional significance is not fully understood, it has been shown that it correlates with the involvement of the left cingulate cortex in language tasks in right-handed subjects (Toga and Thompson 2003). In addition, alteration of the leftward asymmetry is observed in psychiatric diseases such as schizophrenia (Yücel et al. 2002, 2003; Meredith et al. 2012).

We added the following paragraph in the Discussion to clarify this point:

“The leftward asymmetry observed in the human brain appears to be a human trait under the influence of genetic factors and the in-womb environment⁹. Although its functional significance is not fully understood, it has been shown that it correlates with the involvement of the left cingulate cortex in language tasks in right-handed subjects⁵⁹.”

5. First sentence in introduction: “Is the human brain unique?” should be “How is the human brain unique?”. All brains are unique in their own ways.

We fixed this sentence.

6. Last paragraph on p. 5 is non sequitur: “Opposing views suggest that the MFC is functionally comparable in human and macaque”. They can’t be opposing if they suggest the same thing.

We modified this sentence. It now reads:

“Whether the MFC is functionally comparable in human and macaque is subject to debate^{45,46}”

Reviewers' Comments:

Reviewer #1:

Remarks to the Author:

I congratulate the authors for their excellent paper.
I have no further comments.

Reviewer #2:

Remarks to the Author:

The authors have done an excellent job at addressing my concerns from the previous review round.

Reviewer #3:

Remarks to the Author:

The authors have addressed my points as well as they could, and I find the more extensive discussion useful and informative. Thank you.